# Recursive Causal Structure Learning in the Presence of Latent Variables and Selection Bias

**Sina Akbari**
Department of Computer and
Communication Sciences
EPFL, Lausanne, Switzerland
`sina.akbari@epfl.ch`

**Ehsan Mokhtarian**
Department of Computer and
Communication Sciences
EPFL, Lausanne, Switzerland
`ehsan.mokhtarian@epfl.ch`

**AmirEmad Ghassami**
Department of Computer Science
Johns Hopkins University, Baltimore, USA
`aghassa1@jhu.edu`

**Negar Kiyavash**
College of Management of Technology
EPFL, Lausanne, Switzerland
`negar.kiyavash@epfl.ch`

## Abstract

We consider the problem of learning the causal MAG of a system from observational data in the presence of latent variables and selection bias. Constraint-based methods are one of the main approaches for solving this problem, but the existing methods are either computationally impractical when dealing with large graphs or lacking completeness guarantees. We propose a novel computationally efficient recursive constraint-based method that is sound and complete. The key idea of our approach is that at each iteration a specific type of variable is identified and removed. This allows us to learn the structure efficiently and recursively, as this technique reduces both the number of required conditional independence (CI) tests and the size of the conditioning sets. The former substantially reduces the computational complexity, while the latter results in more reliable CI tests. We provide an upper bound on the number of required CI tests in the worst case. To the best of our knowledge, this is the tightest bound in the literature. We further provide a lower bound on the number of CI tests required by any constraint-based method. The upper bound of our proposed approach and the lower bound at most differ by a factor equal to the number of variables in the worst case. We provide experimental results to compare the proposed approach with the state of the art on both synthetic and real-world structures.

## 1 Introduction

Learning the causal structure among the set of variables in the system is the initial step for performing statistical inference tasks such as estimating the reward of a policy in off-policy evaluation [23, 9, 12], etc. In the literature, structure learning is for the most part done under the assumption that all the variables in the system are observed [21, 10, 17, 11, 25]. However, in many applications in real-life systems, this assumption is violated. Moreover, the accessible data may contain selection bias, i.e., some of the variables may have been conditioned on.

The problem of causal structure learning is significantly more challenging when unmeasured (latent) confounders and selection variables exist in the system. This is because the set of directed acyclic graphs (DAGs) as independence models, which is the predominant modeling approach in the absence of unobserved variables, is not closed under marginalization and conditioning [19]. That is, there does not necessarily exist a DAG over the observed variables that demonstrate a one-to-one map with

35th Conference on Neural Information Processing Systems (NeurIPS 2021).

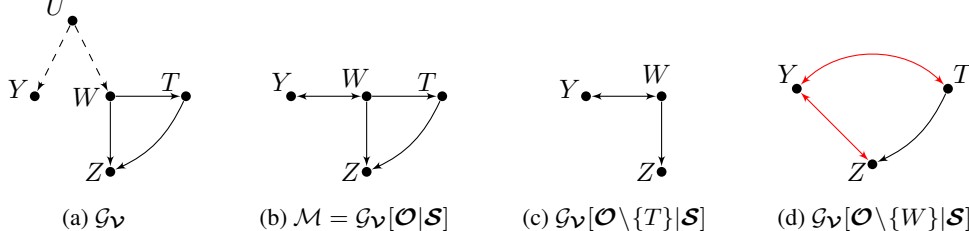

(a) $\mathcal{G}_{\mathcal{V}}$      (b) $\mathcal{M} = \mathcal{G}_{\mathcal{V}}[\mathcal{O}|\mathcal{S}]$      (c) $\mathcal{G}_{\mathcal{V}}[\mathcal{O}\backslash\{T\}|\mathcal{S}]$      (d) $\mathcal{G}_{\mathcal{V}}[\mathcal{O}\backslash\{W\}|\mathcal{S}]$

Figure 1: Effect of removing a variable on the MAG over the remaining variables.

the conditional independence relationships in the observational distribution $P_{\mathcal{O}|\mathcal{S}}$, where $\mathcal{O}$ and $\mathcal{S}$ denote the observed variables and selection variables, respectively. To address this problem, several extensions of the DAG models, such as acyclic directed mixed graphs (ADMGs) [18], induced path graphs (IPGs) [21], and maximal ancestral graphs (MAGs) [19] are introduced in the literature.

The main approaches for structure learning include constraint-based and score-based methods [21, 27, 13, 4]. There are also methods that require specific assumptions on the data generating modules, such as requiring linearity[30], linearity and non-Gaussianity of the noises [20] or additivity of the noise with specific types of non-linearity [8] (See [32] for a summary of structure learning approaches.) Constraint-based methods are the most commonly used methods for structure learning in the presence of latent variables and selection bias [21, 24, 4, 22, 16]. The main idea in these methods is to find the structure which is most consistent with the conditional independence (CI) relationships in the data [21]. However, the sheer number of CI tests required by these methods prohibits applying them to systems with large number of variables.

Several methods are proposed in the literature to reduce the number of CI tests needed in constraint-based methods, specifically when there are no latent and selection variables in the system. For instance, [21] proposed the seminal PC algorithm for graphs with bounded degree, which has polynomial complexity in the number of vertices. [10, 17, 11] proposed using Markov boundary information to reduce the number of required CI tests. If the size of the Markov boundaries or the in-degree of the variables is bounded, these methods achieve quadratic complexity in the number of the variables. However, the majority of the work on causal structure learning in the presence of latent and selection variables do not provide any analysis for the required number of CI tests. As an exception, for sparse graphs, and given the exact value of the maximum degree of the MAG as side information, [3] proposed an algorithm that requires a polynomial number of CI tests in the number of variables. Additionally, [4] proposed a modification of the FCI algorithm, called RFCI, with specific attention to its time complexity. However, RFCI is not complete; that is, the output of this algorithm does not capture all the CI relationships in the data.

In this paper, we propose a novel recursive constraint-based method for causal structure learning in the presence of latent confounders and selection bias. We use MAGs as the graphical representation of the system. The main idea of our recursive approach is that in each iteration, we choose a particular variable of the system, say $X$, and locally identify its adjacent variables. Then, we recursively learn the structure over the rest of the variables using the marginal distribution $P_{\mathcal{O}\backslash\{X\}|\mathcal{S}}$. Note that the choice of $X$ cannot be arbitrary. For instance, consider the DAG $\mathcal{G}_{\mathcal{V}}$ in Figure 1a, where the variable $U$ is latent. The causal MAG over $\mathcal{O} = \{T, W, Y, Z\}$ is shown in Figure 1b as MAG $\mathcal{M}$. As seen in Figure 1c, if we start with the choice of $X = T$, we can correctly learn the subgraph of $\mathcal{M}$ over $\{Y, W, Z\}$, whereas if we start with $X = W$, we will end up learning the graph in Figure 1d, which has two extra edges (highlighted in red) between $Y, Z$ and $Y, T$ that do not exist in $\mathcal{M}$ (we will revisit this example in Section 3).

Our main contributions are as follows.

- We introduce the notion of a *removable* variable in MAGs, which is a variable that can be removed from the causal graph without changing the m-separation relations (Definition 4). We further represent a method to test the removability of a variable given the observational data (Theorem 2).

- We propose an algorithm called L-MARVEL for causal structure learning in the presence of latent and selection variables. We show that our method is sound and complete (Theorem 3) and performs $\mathcal{O}(n^2 + n\Delta_{in}^+(\mathcal{M})^2 2^{\Delta_{in}^+(\mathcal{M})})$ CI tests in the worst case (Proposition 3), where $n$ denotes the number

of variables and $\Delta_{\mathrm{in}}^{+}(\mathcal{M})$ is the maximum size of the union of parents, district and the parents of district of a vertex in the MAG (Equation (5)).

- We show that any constraint-based algorithm requires $\Omega(n^2 + n\Delta_{\mathrm{in}}^{+}(\mathcal{M})2^{\Delta_{\mathrm{in}}^{+}(\mathcal{M})})$ CI tests in the worst case (Theorem 4). Comparing this lower bound with our upper bound demonstrates the efficiency of our proposed method.

To sum up, the purpose and desirability of the proposed recursive algorithm for structure learning is two fold. First, since we choose specific (*removable*) variables in each iteration (with the property of having small Markov boundary), we ensure that the number of required CI tests in each iteration, and hence in total, remains small. Therefore, we can significantly reduce the time complexity compared to non-recursive approaches. Second, by virtue of the gradual reduction of the order of the graph over the iterations, the size of the conditioning sets used in each CI test is reduced, which results in more reliable CI tests with smaller errors and more accurate results.

This paper is organized as follows. In Section 2, we review the preliminaries, present the terminology, and formally describe the problem. In Section 3, we present the L-MARVEL method along with its analysis. In Section 3.3 we also provide the universal lower bound on the complexity of every constraint-based method. Finally, Section 4 presents a comprehensive set of experiments to compare L-MARVEL with various algorithms on synthetic and real-world structures.

## 2 Preliminaries and problem description

### 2.1 Terminology

A *mixed graph* $\mathcal{G}$ over the set of vertices $\mathbf{V}$ is a graph containing three types of edges $-, \rightarrow$ and $\leftrightarrow$. The two ends of an edge are called *marks*. There are two kinds of marks: *arrowhead* ($>$) and *tail* ($-$). If there exists a *directed edge* $X \rightarrow Y$ in the graph, we say $X$ is a *parent* of $Y$ and $Y$ is a *child* of $X$. For a *bi-directed edge* $X \leftrightarrow Y$, we say $X$ and $Y$ are *spouses*. For an *undirected edge* $X - Y$, $X$ and $Y$ are called *neighbors*. In all of the aforementioned cases, we say $X$ and $Y$ are *adjacent*. The *skeleton* of $\mathcal{G}$ is an undirected graph with the same set of vertices $\mathbf{V}$ where there is an edge between $X$ and $Y$ if they are adjacent in $\mathcal{G}$. A path from $X$ to $Y$ where every vertex on the path is a child of its preceding vertex is called a *directed path*. If a directed path exists from $X$ to $Y$, $X$ is called an *ancestor* of $Y$. We assume every vertex is an ancestor of itself. We denote by $Pa(X)$, $Ch(X)$, $N(X)$, $Adj(X)$, and $Anc(X)$, the set of parents, children, neighbors, adjacent vertices, and ancestors of $X$, respectively. The *district set* of a variable $X$, denoted by $Dis(X)$, is the set of variables that have a path to $X$ comprised of only bidirectional edges. By $Pa^{+}(X)$ we denote the union of parents, district set, parents of district set, and the neighbors of a variable[1], i.e.,

$$Pa^{+}(X) = Pa(X) \cup Dis(X) \cup Pa(Dis(X)) \cup N(X). \tag{1}$$

Uppercase capitals indicate single vertices, whereas bold letters denote sets of vertices. For a set of vertices $\mathbf{X}$, $Anc(\mathbf{X}) = \cup_{X \in \mathbf{X}} Anc(X)$. A non-endpoint vertex $X$ on a path is called a *collider*, if both of the edges incident to $X$ on the path have an arrowhead at $X$. A path $\mathcal{P}$ is a *collider path* if every non-endpoint vertex on $\mathcal{P}$ is a collider on $\mathcal{P}$. A path $\mathcal{P}$ between the vertices $X$ and $Y$ is called an *m-connecting* or *active* path relative to a set $\mathbf{Z} \subseteq \mathbf{V} \setminus \{X, Y\}$, if (i) every non-collider on $\mathcal{P}$ is not a member of $\mathbf{Z}$, and (ii) every collider on $\mathcal{P}$ belongs to $Anc(\{X, Y\} \cup \mathbf{Z})$.

**Definition 1** (m-separation). *Suppose $\mathcal{G}$ is a mixed graph. A set $\mathbf{Z}$ m-separates $X$ and $Y$ in $\mathcal{G}$, denoted by $(X \perp Y | \mathbf{Z})_{\mathcal{G}}$, if there is no m-connecting path between $X$ and $Y$ relative to $\mathbf{Z}$ in $\mathcal{G}$[2]. We call $\mathbf{Z}$ a separating set for $X$ and $Y$. We drop the subscript $\mathcal{G}$ whenever it is clear from context.*

A *directed cycle* exists in a mixed graph if $X \rightarrow Y$ and $Y \in Anc(X)$. An *almost directed cycle* exists in a mixed graph when $X \leftrightarrow Y$ and $Y \in Anc(X)$. A mixed graph is said to be *ancestral*, if it does not contain directed cycles or almost-directed cycles, and for any undirected edge $X - Y$, $X$ and $Y$ have no parents or spouses. An ancestral graph is called *maximal* if for any pair of non-adjacent

---

[1]The motivation behind this definition is that the local Markov property does not necessarily hold when causal sufficiency is violated, but if $X \notin Anc(Dis(X))$, then $Pa^{+}(X)$ separates $X$ from its non-descendants. See the supplementary material for proofs.

[2]DAGs are a subclass of mixed graphs. Note that for DAGs, this definition reduces to d-separation. See [14] for the definition of d-separation.

vertices, there exists a set of vertices that m-separates them. A mixed graph is called a *Maximal Ancestral Graph* (MAG) if it is ancestral and maximal. A MAG is called a *directed acyclic graph* (DAG) if it has only directed edges.

A DAG $\mathcal{G}$ can be projected into a unique MAG over a subset of its vertices with the following projection, referred to as an embedded pattern in [28].

**Definition 2** (Latent projection). *Suppose $\mathcal{G}$ is a DAG over $\mathbf{V} = \mathbf{O} \cup \mathbf{L} \cup \mathbf{S}$. The projection of $\mathcal{G}$ over $\mathbf{O}$ conditioned on $\mathbf{S}$, denoted by $\mathcal{G}[\mathbf{O}|\mathbf{S}]$, is a MAG over vertices $\mathbf{O}$ constructed as follows:*

*(i) Skeleton: $X, Y \in \mathbf{O}$ are adjacent in $\mathcal{G}[\mathbf{O}|\mathbf{S}]$ if there exists an inducing path[3] in $\mathcal{G}$ between $X$ and $Y$ relative to $\langle \mathbf{L}, \mathbf{S} \rangle$.*

*(ii) Orientation: For each pair of adjacent variables $X, Y$ in $\mathcal{G}[\mathbf{O}|\mathbf{S}]$, the edge between $X$ and $Y$ is oriented as $X \to Y$ if $X \in Anc(\{Y\} \cup \mathbf{S})$ and $Y \notin Anc(\{X\} \cup \mathbf{S})$; as $X \leftrightarrow Y$ if $X \notin Anc(\{Y\} \cup \mathbf{S})$ and $Y \notin Anc(\{X\} \cup \mathbf{S})$; and as $X - Y$ if $X \in Anc(\{Y\} \cup \mathbf{S})$ and $Y \in Anc(\{X\} \cup \mathbf{S})$.*

The above projection is the unique projection which satisfies the following property [19].

$$(X \perp Y | \mathbf{Z})_{\mathcal{G}[\mathbf{O}|\mathbf{S}]} \iff (X \perp Y | \mathbf{Z} \cup \mathbf{S})_{\mathcal{G}}. \tag{2}$$

Two MAGs are called *Markov equivalent* if they impose the same m-separations. A class of Markov equivalent MAGs can be represented as a (maximally informative) *partially-oriented ancestral graph* (PAG), where the PAG contains the skeleton and all the invariant edge marks in the class.

Let $P$ be the joint distribution over a set of variables $\mathbf{V}$. For $X, Y \in \mathbf{V}, \mathbf{Z} \subseteq \mathbf{V} \backslash \{X, Y\}$, a conditional independence (CI) test in $P$ on the triplet $\langle X, \mathbf{Z}, Y \rangle$ yields independence, denoted by $(X \perp\!\!\!\perp Y | \mathbf{Z})_P$, if $P(X|Y, \mathbf{Z}) = P(X|\mathbf{Z})$. We drop the subscript $P$ when it is clear from context. Suppose $\mathcal{G}$ is a DAG over $\mathbf{V}$, i.e., each vertex of $\mathcal{G}$ corresponds to a variable of $\mathbf{V}$. We say $P$ is *faithful* with respect to $\mathcal{G}$, if $(X \perp\!\!\!\perp Y | \mathbf{Z})_P \iff (X \perp Y | \mathbf{Z})_{\mathcal{G}}$, i.e., the conditional independence in distribution $P$ is equivalent to m-separation in the DAG $\mathcal{G}$.

## 2.2 Problem description

We consider a system with the set of variables $\mathcal{V} = \mathcal{O} \cup \mathcal{L} \cup \mathcal{S}$ and the joint distribution $P_{\mathcal{V}}$, where $\mathcal{O}, \mathcal{L}$, and $\mathcal{S}$ denote the set of observed, latent, and selection variables, respectively. Each variable $X \in \mathcal{V}$ is generated as $X = f_X(Pa(X), \epsilon_X)$, where $f_X$ is a deterministic function, $Pa(X) \subseteq \mathcal{V} \backslash \{X\}$ is the set of parents of $X$, i.e., the set of variables that have a direct causal effect on $X$, and $\epsilon_X$ is the exogenous noise corresponding to $X$. We assume all noise variables are jointly independent. This model is referred to as structural equations model (SEM) [15]. The causal graph of the system, which represents the causal relations among the variables, is denoted by $\mathcal{G}_{\mathcal{V}}$. $\mathcal{G}_{\mathcal{V}}$ is a directed graph over $\mathcal{V}$, i.e., each vertex is associated with a variable[4], and a directed edge exists from each variable in $Pa(X)$ to $X$, for all $X \in \mathcal{V}$. We assume that $\mathcal{G}_{\mathcal{V}}$ is a DAG, and its latent projection over $\mathcal{O}$ conditioned on $\mathcal{S}$ is denoted by $\mathcal{M} := \mathcal{G}_{\mathcal{V}}[\mathcal{O}|\mathcal{S}]$. We will call $\mathcal{M}$ the ground truth MAG. Further, we assume that $P_{\mathcal{V}}$ is faithful with respect to $\mathcal{G}_{\mathcal{V}}$, which along with Equation (2) implies that for each $X, Y \in \mathcal{O}$ and $\mathbf{Z} \subseteq \mathcal{O} \backslash \{X, Y\}$,

$$(X \perp Y | \mathbf{Z})_{\mathcal{M}} \iff (X \perp\!\!\!\perp Y | \mathbf{Z})_{P_{\mathcal{O}|\mathcal{S}}}. \tag{3}$$

Given the observational data from $P_{\mathcal{O}|\mathcal{S}}$, i.e., the marginal distribution over the observed variables, conditioned on the selection variables, we consider the problem of learning the PAG that represents the Markov equivalence class (MEC) of $\mathcal{M}$.

## 3 L-MARVEL Algorithm

In this section, we present *Latent MARVEL* (L-MARVEL) algorithm to learn the PAG over $\mathcal{O}$ that represents the system. This algorithm relies on a notion similar to the MARVEL algorithm proposed by [11] for DAG learning when all the variables are observable. Our approach relies on the Markov boundary information as input.

---

[3]An *inducing path* between $X$ and $Y$ relative to $\langle \mathbf{L}, \mathbf{S} \rangle$, where $\mathbf{L}$ and $\mathbf{S}$ are disjoint sets not containing $X$ and $Y$, is a path on which every non-collider is a member of $\mathbf{L}$ and every collider belongs to $Anc(\{X, Y\} \cup \mathbf{S})$.

[4]We will use vertex and variable interchangeably throughout the paper.

**Algorithm 1:** L-MARVEL.

1: **Input:** $\mathcal{O}$, $P_{\mathcal{O}|\boldsymbol{\mathcal{S}}}$
2: **Output:** PAG $\hat{\mathcal{M}}$
3: $Mb_{\mathcal{O}} \leftarrow$ ComputeMb($\mathcal{O}$, $P_{\mathcal{O}|\boldsymbol{\mathcal{S}}}$)
4: $\mathcal{A} \leftarrow$ Initialization($\mathcal{O}$, $Mb_{\mathcal{O}}$)
5: $\mathcal{A} \leftarrow$ L-MARVEL($\mathcal{O}$, $P_{\mathcal{O}|\boldsymbol{\mathcal{S}}}$, $Mb_{\mathcal{O}}$, $\mathcal{A}$)
6: Create $\hat{\mathcal{M}}$ according to adjacencies in $\mathcal{A}$ and orient it maximally using rules 0-10 of [31]

---

1: **Function** L-MARVEL($\mathbf{V}$, $P_{\mathbf{V}|\boldsymbol{\mathcal{S}}}$, $Mb_{\mathbf{V}}$, $\mathcal{A}$)
2: **if** $|\mathbf{V}| = 1$ **then**
3:     **return** $\mathcal{A}$
4: **else**
5:     $(X_1, X_2, \ldots X_{|\mathbf{V}|}) \leftarrow$ Sort $\mathbf{V}$ based on the Markov boundary size in ascending order.
6:     **for** $i = 1$ to $|\mathbf{V}|$ **do**
7:         $(Adj(X_i), \mathcal{A}) \leftarrow$ **FindAdjacent**($X_i$, $Mb_{\mathbf{V}}(X_i)$, $P_{\mathbf{V}|\boldsymbol{\mathcal{S}}}$, $\mathcal{A}$)
8:         $isR \leftarrow$ **IsRemovable**($X_i$, $Mb_{\mathbf{V}}(X_i)$, $P_{\mathbf{V}|\boldsymbol{\mathcal{S}}}$, $Adj(X_i)$)     % Main step of the algorithm.
9:         **if** $isR$ is true **then**
10:             $(Mb_{\mathbf{V}\backslash X_i}, \mathcal{A}) \leftarrow$ **UpdateMb**($X_i$, $Adj(X_i)$, $Mb_{\mathbf{V}}$, $P_{\mathbf{V}|\boldsymbol{\mathcal{S}}}$, $\mathcal{A}$)
11:             **return** L-MARVEL($\mathbf{V}\backslash\{X_i\}$, $P_{\mathbf{V}\backslash\{X_i\}|\boldsymbol{\mathcal{S}}}$, $Mb_{\mathbf{V}\backslash\{X_i\}}$, $\mathcal{A}$)

---

**Definition 3** (Markov boundary). *Suppose* $\mathbf{V} \subseteq \mathcal{O}$*. Markov boundary of* $X \in \mathbf{V}$ *with respect to* $\mathbf{V}$ *is a minimal set of variables* $\mathbf{Z} \subseteq \mathbf{V}\backslash\{X\}$*, such that* $X$ *is independent of the rest of the variables of* $\mathbf{V}$ *conditioned on* $\mathbf{Z} \cup \boldsymbol{\mathcal{S}}$*.*

Under faithfulness, Markov boundary of $X \in \mathbf{V} \subseteq \mathcal{O}$ with respect to $\mathbf{V}$, denoted by $Mb_{\mathbf{V}}(X)$, is unique and it consists of all the variables that have a collider path to $X$ in $\mathcal{G}_{\boldsymbol{\mathcal{V}}}[\mathbf{V}|\boldsymbol{\mathcal{S}}]$ [29, 16]. We indicate by $Mb_{\mathbf{V}}$, the Markov boundaries of all of the variables in $\mathbf{V}$ with respect to $\mathbf{V}$.

Our learning procedure is outlined in Algorithm 1. Throughout the algorithm, the data structure $\mathcal{A}$ stores the pairs of vertices that have been identified to be adjacent, and the separating sets found for non-adjacent vertices so far. As the first step, the Markov boundary information with respect to $\mathcal{O}$ is identified using one of the standard methods in the literature, as discussed in Section 3.2. Then $\mathcal{A}$ is initialized with separating sets implied by the Markov boundary information, i.e., for any $X$ and $Y \notin Mb_{\mathcal{O}}(X)$, $Mb_{\mathcal{O}}(X)$ is a separating set for $X, Y$. $\mathcal{A}$ is updated when a new separating set is discovered for a pair of vertices, or two vertices are determined to be adjacent. After initializing $\mathcal{A}$ in line 4, we call the L-MARVEL function over $\mathcal{O}$, which recursively identifies all the adjacent pairs of vertices, i.e., the skeleton of $\mathcal{M}$, and discovers a separating set for all non-adjacent pairs of vertices. This information suffices to maximally orient the edge marks at the end of the algorithm using the complete set of orientation rules in [31].

L-MARVEL works as follows. It chooses a variable $X$, identifies $Adj(X)$ (i.e., the set of variables adjacent to $X$), and then recursively learns the structure over $\mathcal{O}\backslash\{X\}$, discarding $X$. This is desirable as the problem size decreases at each iteration, which results in a substantial reduction in the computational complexity. Moreover, performing CI tests of high order is avoided. If the learned graph, i.e., $\mathcal{G}_{\boldsymbol{\mathcal{V}}}[\mathcal{O}\backslash\{X\}|\boldsymbol{\mathcal{S}}]$ is equal to the induced subgraph of $\mathcal{G}_{\boldsymbol{\mathcal{V}}}[\mathcal{O}|\boldsymbol{\mathcal{S}}]$ over $\mathcal{O}\backslash\{X\}$, we can add $X$ to this graph and connect it to its adjacent variables with an edge. As discussed in the example in Figure 1, this is not true for an arbitrary $X$. We show that for certain vertices, called *removable*, we can indeed apply such a recursive learning procedure. Next, we define what makes a variable removable.

**Definition 4** (Removable). *Suppose* $\mathcal{G}$ *is a MAG over* $\mathbf{V}$*,* $X \in \mathbf{V}$*, and* $\mathcal{H}$ *is the induced subgraph of* $\mathcal{G}$ *over* $\mathbf{V}\backslash\{X\}$*.* $X$ *is a removable vertex in* $\mathcal{G}$ *if* $\mathcal{G}$ *and* $\mathcal{H}$ *impose the same m-separation relations over* $\mathbf{V}\backslash\{X\}$*. That is, for any vertices* $Y, W \in \mathbf{V}\backslash\{X\}$ *and* $\mathbf{Z} \subseteq \mathbf{V}\backslash\{X, Y, W\}$*,*

$$(Y \perp W | \mathbf{Z})_{\mathcal{G}} \iff (Y \perp W | \mathbf{Z})_{\mathcal{H}}. \tag{4}$$

In the case that $\mathcal{G}$ is a DAG, Definition 4 reduces to what [11] proposed for DAGs. However, their tests for identifying removability fail when causal sufficiency is violated. Next, we provide a graphical characterization of removable variables.

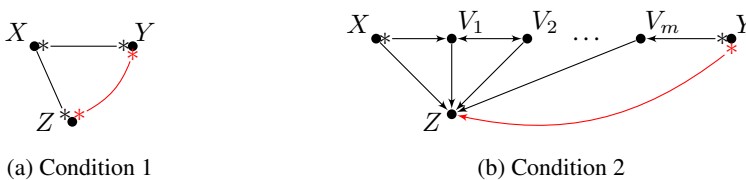

(a) Condition 1                               (b) Condition 2

Figure 2: Graphical characterization of a removable variable. The edge marks indicated by a star ($*$) can be either a tail or an arrowhead.

**Theorem 1.** *Vertex $X$ is removable in a MAG $\mathcal{M}$ over the variables $\mathbf{V}$, if and only if*

1. *for any $Y \in Adj(X)$ and $Z \in Ch(X) \cup N(X) \setminus \{Y\}$, $Y$ and $Z$ are adjacent, and*

2. *for any collider path $u = (X, V_1, ..., V_m, Y)$ and $Z \in \mathbf{V} \setminus \{X, Y, V_1, ..., V_m\}$ such that $\{X, V_1, ..., V_m\} \subseteq Pa(Z)$, $Y$ and $Z$ are adjacent.*

Figure 2 represents the graphical constraints of Theorem 1. Figure 2a depicts the first condition, where $Z$ is either a child or a neighbor of $X$, and $Y \in Adj(X)$, while Figure 2b depicts a collider path where $X$ and $V_i$s are parents of $Z$. Theorem 1 states that $X$ is removable if and only if the edges highlighted in red are present in both cases. See Appendix A for a formal proof and further discussion on Theorem 1. The next proposition clarifies why removable variables are exactly those that can be removed at each iteration in our recursive approach.

**Proposition 1.** *Suppose $\mathbf{V} \subseteq \mathcal{O}$ and $X \in \mathbf{V}$. $\mathcal{G}_{\mathcal{V}}[\mathbf{V} \setminus \{X\} | \mathcal{S}]$ is equal to the induced subgraph of $\mathcal{G}_{\mathcal{V}}[\mathbf{V} | \mathcal{S}]$ over $\mathbf{V} \setminus \{X\}$ if and only if $X$ is removable in $\mathcal{G}_{\mathcal{V}}[\mathbf{V} | \mathcal{S}]$.*

Appendix B includes the proofs of our results. Identifying a removable variable at each iteration is the core of L-MARVEL. We will discuss an efficient algorithm to determine whether a variable is removable in Section 3.1. At each iteration, given the set of remaining variables $\mathbf{V}$, these variables are sorted in ascending order of their Markov boundary size. Starting with the variable with the smallest Markov boundary, we search for its adjacent vertices within its Markov boundary. If $Y \in Mb_{\mathbf{V}}(X_i)$ is not adjacent to $X_i$, then $X_i$ and $Y$ have a separating set in $Mb_{\mathbf{V}}(X_i) \setminus \{Y\}$ [16]. Hence, identifying $Adj(X_i)$ can be performed using a brute-force search in the Markov boundary, using at most $|Mb_{\mathbf{V}}(X_i)| 2^{(|Mb_{\mathbf{V}}(X_i)|-1)}$ CI tests. In Section 3.3, we show that the loop in line 6 of Algorithm 1 never reaches variables with large Markov boundaries, and this guarantees that both the number of CI tests and their order remains small throughout the algorithm. We then determine whether $X_i$ is removable given $Mb_{\mathbf{V}}(X_i)$ and $Adj(X_i)$, using the efficient algorithm we shall discuss in Section 3.1. We continue this procedure until we identify the first removable variable $X = X_i$. Then, we remove $X$ from the set of remaining variables, and update the Markov boundaries with respect to $\mathbf{V} \setminus \{X\}$, which is the input to the next iteration. The latter does not require the discovery of Markov boundaries from scratch, and is implemented as we shall see in Section 3.2.

The rest of the section is dedicated to showing how to efficiently identify a *removable* variable (Section 3.1), how to update the Markov boundary information (Section 3.2), and the analysis of the algorithm (Section 3.3).

## 3.1 Testing removability in MAGs

The following theorem presents the conditions of removability of a variable using CI tests within the Markov boundary. This theorem excludes a particular structure of the MAG $\mathcal{M}$, where $\mathcal{M}$ has a cycle of the length of at least four that contains only undirected edges, and this cycle has no chords. We shall discuss in Appendix D, which specific structure of the DAG $\mathcal{G}_{\mathcal{V}}$ this MAG represents, and why it is required to exclude this specific structure. As we shall see in Appendix D, such MAGs imply a very restrictive structure over the selection variables.

**Theorem 2.** *Suppose the edge-induced subgraph of $\mathcal{M}$ over the undirected edges (i.e., the edges due to selection bias) is chordal. Let $\mathcal{G} = \mathcal{G}_{\mathcal{V}}[\mathbf{V} | \mathcal{S}]$ for some $\mathbf{V} \subseteq \mathcal{O}$. $X \in \mathbf{V}$ is removable in $\mathcal{G}$ if and only if for every $Y \in Adj(X)$ and $Z \in Mb_{\mathbf{V}}(X)$, at least one of the following holds.*

***Condition 1:*** *$\exists \mathbf{W} \subseteq Mb_{\mathbf{V}}(X) \setminus \{Y, Z\} \colon Y \perp\!\!\!\perp Z | \mathbf{W}$.*

*Condition 2:* $\forall \mathbf{W} \subseteq Mb_{\mathbf{V}}(X) \backslash \{Y, Z\} \colon Y \not\perp Z | \mathbf{W} \cup \{X\}$.

*Furthermore, the set of removable vertices in $\mathcal{G}$ is non-empty.*

Using Theorem 2 and given $Adj(X)$ and $Mb(X)$, Algorithm 2 tests the removability of $X$ by performing $\mathcal{O}(|Adj(X)||Mb_{\mathbf{V}}(X)|\, 2^{|Mb_{\mathbf{V}}(X)|})$ CI tests. Note that the removability test is only performed for variables with small Markov boundaries, which keeps both the number of CI tests and the size of the conditioning sets small, as we shall see in Section 3.3.

---

**Algorithm 2:** IsRemovable - Determine whether $X$ is removable.

---

1: **Input:** $(X, Mb_{\mathbf{V}}(X), P_{\mathbf{V}|\mathcal{S}}, Adj(X))$
2: **for** $Y \in Adj(X), Z \in Mb_{\mathbf{V}}(X)$ **do**
3:    **if** Condition 1 of Theorem 2 does not holds **then**
4:       **if** Condition 2 of Theorem 2 does not holds **then**
5:          **Return** False           % $X$ is not removable.
6: **Return** True                      % $X$ is removable.

---

Conditions of Theorem 2 can be checked in different orders, although we have witnessed in our experiments that checking these conditions in the order of Algorithm 2 increases the accuracy.

## 3.2 Markov boundary discovery and updating Markov boundaries

L-MARVEL requires Markov boundary information for initialization. Several algorithms have been proposed in the literature for discovering the Markov boundaries[10, 17, 26, 29]. For instance, TC [17] algorithm states that

$$(X \not\perp Y | \mathbf{V} \backslash \{X, Y\})_{P_{\mathbf{V}|\mathcal{S}}} \iff X \in Mb_{\mathbf{V}}(Y) \text{ and } Y \in Mb_{\mathbf{V}}(X),$$

where $\mathbf{V} \subseteq \mathcal{O}$. Grow-Shrink (GS) algorithm [10] and its modifications, including IAMB and its variants [26] address Markov boundary discovery by performing more CI tests with smaller conditioning sets. These algorithms require a linear number of CI tests in the number of variables to determine the Markov boundary of a certain variable, i.e., quadratic number of CI tests to discover the entire Markov boundaries. However, given the challenging nature of Markov boundary discovery, these algorithms might fail to accurately discover this information in some settings. We need to utilize one of these methods[5] to initially discover the Markov boundaries, but the subsequent update of the boundaries throughout the later iterations is performed within L-MARVEL as we shall next discuss.

**Updating Markov boundaries:** Let $Mb_{\mathbf{V}}$ be the input to an iteration of L-MARVEL where $X$ is identified as removable and we need to learn $Mb_{\mathbf{V}\backslash\{X\}}$. By definition of removability, the latent projection of $\mathcal{G}_{\mathcal{V}}$ over $\mathbf{V}\backslash\{X\}$ is the induced subgraph of $\mathcal{G}_{\mathcal{V}}[\mathbf{V}|\mathcal{S}]$. As a result, removing $X$ has two effects: 1. $X$ is removed from all Markov boundaries, and 2. for $Y, Z \in \mathbf{V}\backslash\{X\}$, if all of the collider paths between $Y$ and $Z$ in $\mathcal{G}_{\mathcal{V}}[\mathbf{V}|\mathcal{S}]$ pass through $X$, then $Y$ and $Z$ must be excluded from each others Markov boundary. Note that in the latter case, $Y, Z \in Mb_{\mathbf{V}}(X)$. The latter update is performed using a single CI test, i.e., $(Y \not\perp Z | Mb_{\mathbf{V}}(Z) \backslash \{X, Y, Z\})$, or equivalently, $(Y \not\perp Z | Mb_{\mathbf{V}}(Y) \backslash \{X, Y, Z\})$. We choose the CI test with the smaller conditioning set among the two. If the dependency does not hold, we remove $Y, Z$ from each other's Markov boundary.

## 3.3 Analysis

First, we state the soundness and completeness of L-MARVEL in the following theorem.

**Theorem 3.** *Suppose the distribution $P_{\mathcal{V}}$ over $\mathcal{V} = \mathcal{O} \cup \mathcal{L} \cup \mathcal{S}$ is faithful to the DAG $\mathcal{G}_{\mathcal{V}}$. If the conditional independence relations among all variables in $\mathcal{O}$ given $\mathcal{S}$ is provided to L-MARVEL, the output of L-MARVEL is the PAG representing the Markov equivalence class of $\mathcal{G}_{\mathcal{V}}[\mathcal{O}|\mathcal{S}]$.*

Let $\Delta_{\text{in}}^{+}(\mathcal{H})$ denote the maximum size of $Pa^{+}(\cdot)$ (defined in (1)) in a MAG $\mathcal{H}$, i.e.,

$$\Delta_{\text{in}}^{+}(\mathcal{H}) = \max_{X \in \mathcal{H}} |Pa^{+}(X)|. \tag{5}$$

---

[5]In our experiments, we used TC.

Next, we provide an upper bound on the size of the Markov boundary of a removable variable.

**Proposition 2.** *If $X$ is a removable variable in MAG $\mathcal{H}$ with vertices $\mathbf{V}$, then $|Mb_{\mathbf{V}}(X)| \leq \Delta_{in}^+(\mathcal{H})$.*

L-MARVEL processes variables in the ascending order of their Markov boundary size at each iteration and stops when the first removable variable is identified. Therefore, Proposition 2 guarantees that all the processed vertices at each iteration have Markov boundaries smaller than $\Delta_{in}^+(\mathcal{G}_{\mathcal{V}}[\mathbf{V}|\mathcal{S}])$, where $\mathbf{V}$ is the set of remaining variables. This number gets smaller during the algorithm, as L-MARVEL keeps only a subgraph of the input. Note that this bound applies to the size of the conditioning sets of CI tests performed in functions **FindAdjacent** and **IsRemovable**, since the conditioning sets are a subset of the Markov boundary. Furthermore, it results in the following upper bound on the number of CI tests.

**Proposition 3.** *The number of conditional independence tests Algorithm 1 performs on a MAG $\mathcal{M}$ of order $n$, in the worst case, is upper bounded by*

$$\mathcal{O}(n^2 + n\Delta_{in}^+(\mathcal{M})^2 2^{\Delta_{in}^+(\mathcal{M})}). \tag{6}$$

The quadratic term in the upper bound of Equation (6) is for initial Markov boundary discovery. Note that algorithms such as GS, TC, IAMB, etc. discover the Markov boundary of each variable requiring only linear number of CI tests in $n$.

To the best of our knowledge, this is the tightest bound in the literature. The following lower bound on all of the constraint based structure learning algorithm demonstrates the efficiency of L-MARVEL.

**Theorem 4.** *The number of conditional independence tests of the form $(X \perp\!\!\!\perp Y | \mathbf{Z})$ required by any constraint-based algorithm on a MAG $\mathcal{M}$ of order $n$, in the worst case, is lower bounded by*

$$\Omega(n^2 + n\Delta_{in}^+(\mathcal{M})2^{\Delta_{in}^+(\mathcal{M})}). \tag{7}$$

Comparing this lower bound with our achievable upper bound, we can see that the complexity of L-MARVEL in the worst case is merely different by a factor which is at most the number of observed variables in the worst case.

## 4 Experiments

We report empirical results on both synthetic (random graphs) and real-world structures available in the Bayesian network repository[6], the benchmark for structure learning in the literature. We evaluate and compare L-MARVEL[7] to various algorithms, namely the constraint-based methods FCI [21], RFCI [4], and MBCS* [16], and the hybrid method M3HC [27] in terms of both computational complexity and accuracy. Following the convention in [4, 27, 16, 2], the data is generated according to a linear SEM with additive Gaussian noise, where all the variables of the system (including latent and selection) are generated as linear combinations of their parents plus a Gaussian noise. For each system, we simulate data from $P_{\mathcal{O}|\mathcal{S}}$, the data available to all the algorithms. We use TC [17] algorithm to learn the initial Markov boundaries. To make a fair comparison among the algorithms, we feed the Markov boundary information to all the algorithms, that is, algorithms start from a graph where the edges between vertices that are not in each other's Markov boundary are already deleted. This is similar to the ideas in [17]. For CI tests, we use Fisher Z-transformation [6] with significance level $\alpha = 0.01$ for all the algorithms, and $\alpha = 2/n^2$ for TC [17]. In all the experiments, each point on the plots and each entry of the table represents an average of 50 datasets, where the latent and selection variables were chosen uniformly at random.

**Random Structures:** We used two different generating processes to obtain our random graphs. 1. MAGs corresponding to DAGs generated by Erdos-Renyi model $G(\tilde{n}, p)$ [5], where $\tilde{n}$ denotes the total number of the variables, and 2. MAGs corresponding to random DAGs where each vertex has a maximum of 3 or 4 parents, similar to the setting in [27, 11, 2]. Figures 3a and 3b illustrate the performance of the algorithms on Erdos Renyi graphs, whereas Figures 3c, 3d and 3e represent the performance of these methods on the latter generative model. The coefficients of the linear model and the standard deviation of the exogenous noises are chosen uniformly at random from

---

[6]bnlearn.com/bnrepository/

[7]The implementation of L-MARVEL is available in github.com/Ehsan-Mokhtarian/L-MARVEL.

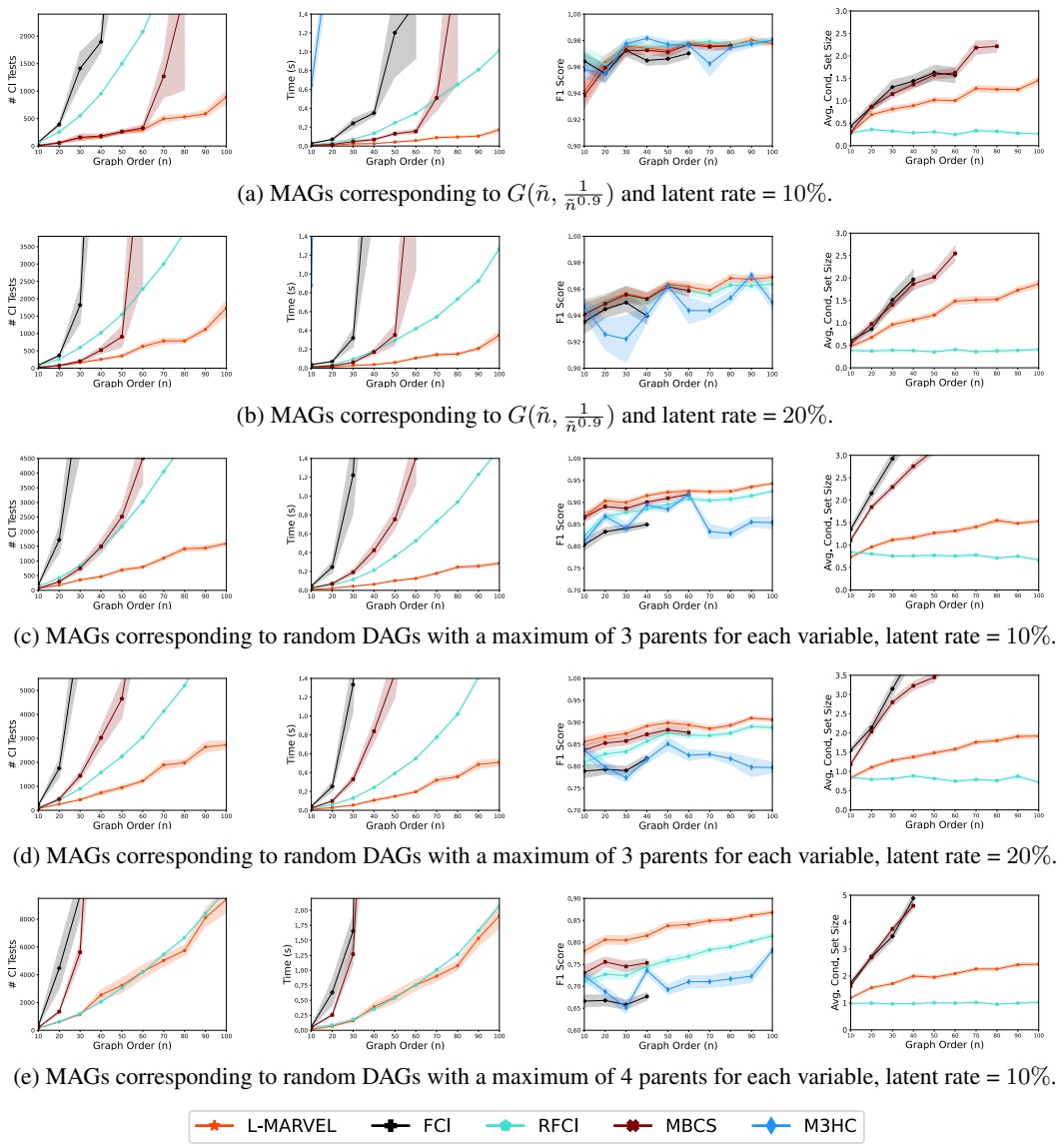

(a) MAGs corresponding to $G(\tilde{n}, \frac{1}{\tilde{n}^{0.9}})$ and latent rate = 10%.

(b) MAGs corresponding to $G(\tilde{n}, \frac{1}{\tilde{n}^{0.9}})$ and latent rate = 20%.

(c) MAGs corresponding to random DAGs with a maximum of 3 parents for each variable, latent rate = 10%.

(d) MAGs corresponding to random DAGs with a maximum of 3 parents for each variable, latent rate = 20%.

(e) MAGs corresponding to random DAGs with a maximum of 4 parents for each variable, latent rate = 10%.

Figure 3: Performance of various algorithms on random graphs with significance level $\alpha = 0.01$ and $50|\mathcal{O}|$ samples are available. Figures (a) and (b) demonstrate the evaluation over MAGs corresponding to Erdos-Renyi graphs, while (c), (d) and (e) represent the MAGs corresponding to random DAGs with bounded number of parents for each variable (sample size = $50|\mathcal{O}|$).

$\pm(0.5, 2)$ and $(1, \sqrt{3})$, respectively. We did not continue running algorithms that were not capable of keeping up with the cohort as the order of the graphs grew. Moreover, the runtime of M3HC is not reported in the plots as it does not fit into the scale of the plots. As seen in the plots, L-MARVEL demonstrates substantially lower computational complexity in terms of number of the CI tests and runtime compared to the other algorithms, while maintaining high accuracy (the highest among the cohort in most of the cases). We also observed a low size for the conditioning sets in our CI tests for L-MARVEL. Only RFCI performs better than L-MARVEL in this metric[8].

**Benchmark Structures:** Algorithms are evaluated on benchmark structures, where 5% to 10% of the variables are assumed to be latent, and $\sim 5\%$ of them are selection variables. Latent and selection variables are chosen uniformly at random for each dataset. The coefficients of the linear model and

---

[8]Note that RFCI avoids performing too many CI tests but with the caveat of lacking completeness.

the standard deviation of the noises are chosen uniformly at random from $\pm(0.5, 1)$ and $(\sqrt{0.5}, 1)$, respectively. Our experiments, summarized in Table 1, demonstrate that L-MARVEL outperforms the other algorithms both in terms of computational complexity and the accuracy of the learned structure. NA entries for FCI demonstrate that the runtime exceeds a certain threshold.

Table 1: Performance of various algorithms on the benchmark structures, when $5\%$ to $10\%$ of the variables are latent and $\sim 5\%$ of them are selection variables (sample size $= 50|\mathcal{O}|$).

| | Structure $(|\mathcal{O}|,|\mathcal{L}|,|\mathcal{S}|)$ | Insurance (22,3,2) | Alarm (31,4,2) | Ecoli70 (40,3,3) | Barley (40,5,3) | Hailfinder (50,3,3) | Carpo (53,4,4) | Arth150 (95,6,6) |
|---|---|---|---|---|---|---|---|---|
| **L-MARVEL** | #CI tests | **272** | **235** | **227** | **894** | **333** | **569** | **1185** |
| | Runtime | **0.03** | **0.04** | **0.05** | **0.16** | **0.07** | **0.12** | **0.36** |
| | F1-score | **0.85** | **0.92** | **0.88** | **0.82** | **0.92** | **0.97** | **0.89** |
| | Precision | 0.97 | 0.98 | 0.97 | 0.98 | 0.98 | 0.99 | 0.99 |
| | Recall | **0.76** | **0.87** | **0.81** | **0.72** | **0.87** | **0.96** | **0.82** |
| **RFCI** | #CI tests | 947 | 981 | 4314 | 2158 | 256754 | 11670 | 2644794 |
| | Runtime | 0.14 | 0.20 | 0.86 | 0.44 | 62.22 | 2.59 | 1047.44 |
| | F1-score | 0.76 | 0.89 | 0.85 | 0.73 | 0.88 | 0.94 | 0.87 |
| | Precision | **0.99** | **1.00** | **1.00** | **1.00** | **1.00** | **1.00** | **1.00** |
| | Recall | 0.63 | 0.81 | 0.74 | 0.58 | 0.79 | 0.89 | 0.77 |
| **FCI** | #CI tests | 7117 | 6899 | 56781 | 117566 | NA | 123198 | NA |
| | Runtime | 1.13 | 1.25 | 13.22 | 25.78 | NA | 31.41 | NA |
| | F1-score | 0.75 | 0.88 | 0.83 | 0.70 | NA | 0.45 | NA |
| | Precision | **0.99** | **1.00** | **1.00** | **1.00** | NA | 0.48 | NA |
| | Recall | 0.61 | 0.80 | 0.72 | 0.54 | NA | 0.42 | NA |
| **MBCS*** | #CI tests | 640 | 335 | 499 | 2649 | 502 | 1221 | 3225 |
| | Runtime | 0.12 | 0.11 | 0.17 | 0.77 | 0.19 | 0.46 | 1.94 |
| | F1-score | 0.80 | 0.90 | 0.86 | 0.76 | 0.89 | 0.96 | 0.86 |
| | Precision | 0.98 | 0.98 | 0.98 | 0.99 | 0.99 | 0.99 | 0.99 |
| | Recall | 0.68 | 0.84 | 0.77 | 0.62 | 0.82 | 0.94 | 0.76 |
| **M3HC** | #CI tests | 896 | 674 | 3033 | 1731 | 139788 | 8354 | 793754 |
| | Runtime | 13.66 | 4.19 | 6.64 | 12.53 | 47.72 | 7.42 | 322.33 |
| | F1-score | 0.75 | 0.87 | 0.84 | 0.71 | 0.86 | 0.92 | 0.84 |
| | Precision | **0.99** | **1.00** | **1.00** | 0.99 | **1.00** | **1.00** | 0.99 |
| | Recall | 0.62 | 0.78 | 0.73 | 0.56 | 0.77 | 0.85 | 0.74 |

More comprehensive experimental results including the effect of the sample size, wider range of latent and selection rates, and assessments on different settings of parameters and structures, along with alternative metrics are reported in Appendix C.

# 5 Concluding Remarks

We proposed a recursive structure learning approach capable of handling latent and selection variables. The recursive technique significantly reduced the number of required CI tests (and hence the time complexity). Also, since the order of the graph becomes smaller over the iterations, the recursive approach reduces the size of the conditioning sets in each CI test, which leads to an improved performance of the tests. We provided an upper bound on the complexity of the proposed method as well as a lower bound for any constraint-based method. The upper bound of our proposed approach and the lower bound at most differ by a factor equal to the number of variables in the worst case, which demonstrates the efficiency of the proposal. We compared the performance of the proposed method with several state-of-the-art approaches on both synthetic and real-world structures. The results showed improvement in both performance and complexity on almost all the setups. We note that the performance of the proposed method is reliant on the accuracy of the Markov boundary information that is used in the algorithm. Devising efficient and high accuracy approaches for learning the Markov boundary of the variables is left as an important direction for future work.

## Acknowledgments and Disclosure of Funding

The work presented in this paper was in part supported by Office of Naval Research (ONR) under grant number W911NF-15-1-0479.

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
