# Appendix

## A  Removable Variables

In this section, we first prove the proposed graphical representation for a removable variable in a MAG $\mathcal{M}$ (Theorem 1). Then, we discuss how this representation reduces to Theorem 5 of [11] in the case of DAGs.

Throughout our proofs, we say a path between $X$ and $Y$ is *blocked* by a set $\mathbf{W}$ if it is not m-connecting relative to $\mathbf{W}$. In this case, there exists a non-collider $W$ on the path which is a member of $\mathbf{W}$, or there exists a collider $W$ on the path such that $W \notin Anc(\{X, Y\} \cup \mathbf{W})$. In both cases we say $W$ blocks this path with respect to $\mathbf{W}$, or $W$ blocks the path in short when $\mathbf{W}$ is clear from the context. We say $X$ is a descendant of $Y$ if $Y \in Anc(X)$, and we denote by $De_{\mathcal{M}}(X)$ the set of descendants of $X$ in the MAG $\mathcal{M}$, and $De(X)$ whenever the graph is clear from the context.

### A.1  Graphical representation

**Theorem 1.** *Vertex $X$ is removable in a MAG $\mathcal{M}$ over the variables $\mathbf{V}$, if and only if*

1. *for any $Y \in Adj(X)$ and $Z \in Ch(X) \cup N(X) \setminus \{Y\}$, $Y$ and $Z$ are adjacent, and*

2. *for any collider path $u = (X, V_1, ..., V_m, Y)$ and $Z \in \mathbf{V} \setminus \{X, Y, V_1, ..., V_m\}$ such that $\{X, V_1, ..., V_m\} \subseteq Pa(Z)$, $Y$ and $Z$ are adjacent.*

*Proof.* Let $\mathcal{H}$ denote the induced subgraph of $\mathcal{M}$ over $\mathbf{V} \setminus \{X\}$.

**only if part:** Suppose $Y \in Adj(X)$ and $Z \in Ch(X) \cup N(X)$. For any $\mathbf{W} \subseteq \mathbf{V} \setminus \{X, Y, Z\}$, $(Z, X, Y)$ is an m-connecting path relative to $\mathbf{W}$ in $\mathcal{M}$, as $X$ is a non-collider and $X \notin \mathbf{W}$. That is, no such $\mathbf{W}$ can m-separate $Y$ and $Z$. Since $X$ is removable in $\mathcal{M}$, by definition of removability,

$$(Y \perp Z | \mathbf{W})_{\mathcal{M}} \iff (Y \perp Z | \mathbf{W})_{\mathcal{H}}. \tag{8}$$

As a result, $Y$ and $Z$ have no m-separating sets in $\mathcal{H}$. Hence, $Y$ is adjacent to $Z$ in $\mathcal{H}$, and therefore, in $\mathcal{M}$.

Now suppose $u = (X, V_1, ..., V_m, Y)$ is a collider path and $\{X, V_1, ..., V_m\} \subseteq Pa(Z)$. Again for any $\mathbf{W} \subseteq \mathbf{V} \setminus \{X, Y, Z\}$, $(Z, X, V_1, ..., V_m, Y)$ is an m-connecting path relative to $\mathbf{W}$ in $\mathcal{M}$ since I) every collider on this path is a parent (and therefore an ancestor) of $Z$, and II) $X \notin \mathbf{W}$ and $X$ is the only non-collider on this path. That is, no such $\mathbf{W}$ can m-separate $Y$ and $Z$. Since $X$ is removable in $\mathcal{M}$, Equation 8 implies that $Y$ and $Z$ have no m-separating sets in $\mathcal{H}$. Hence, $Y$ is adjacent to $Z$ in $\mathcal{H}$, and therefore, in $\mathcal{M}$.

**if part:** We need to prove that for any $Y, Z \in \mathbf{V} \setminus \{X\}$ and any $\mathbf{W} \subseteq \mathbf{V} \setminus \{X, Y, Z\}$,

$$(Y \perp Z | \mathbf{W})_{\mathcal{M}} \iff (Y \perp Z | \mathbf{W})_{\mathcal{H}}.$$

$\Rightarrow$: Suppose $(Y \perp Z | \mathbf{W})_{\mathcal{M}}$ and let $u$ be an arbitrary path in $\mathcal{H}$ between $Y$ and $Z$. Since $\mathcal{H}$ is a subgraph of $\mathcal{M}$, $u$ is also a path in $\mathcal{M}$. As $(Y \perp Z | \mathbf{W})_{\mathcal{M}}$, $u$ is not m-connecting relative to $\mathbf{W}$ in $\mathcal{M}$, Lemma 6 implies that $u$ is not m-connecting relative to $\mathbf{W}$ in $\mathcal{H}$.

$\Leftarrow$: Suppose $(Y \perp Z | \mathbf{W})_{\mathcal{H}}$, i.e., there is no m-connecting path between $Y$ and $Z$ in $\mathcal{H}$. It suffices to show that none of the paths between $Y$ and $Z$ in $\mathcal{M}$ are m-connecting. Take an arbitrary path $u = (Y, V_1, ..., V_m, Z)$ in $\mathcal{M}$. We will show that $u$ is not m-connecting relative to $\mathbf{W}$ in $\mathcal{M}$. We consider the following cases separately.

1. $X \notin u$: In this case, $u$ is also a path in $\mathcal{H}$. Since $u$ is not m-connecting relative to $\mathbf{W}$ in $\mathcal{H}$, Lemma 6 implies that $u$ is not m-connecting relative to $\mathbf{W}$ in $\mathcal{M}$.

2. $X$ is a non-collider on $u$: Suppose $u = (Y, V_1, \ldots, V_{i-1}, V_i = X, V_{i+1}, \ldots, V_m, Z)$. We claim that a vertex other than $X$ blocks $u$ in $\mathcal{M}$. Suppose not. Since $X$ is a non-collider, at least one of $V_{i-1}$ and $V_{i+1}$ is a child or neighbor of $X$. From the assumption of the theorem, $V_{i-1} \in Adj(V_{i+1})$. Now consider the path $u' = (Y, V1, ..., V_{i-1}, V_{i+1}, ..., V_m, Z)$, which is a path in $\mathcal{H}$ and must not be m-connecting relative to $\mathbf{W}$ in $\mathcal{H}$. Hence, Lemma 6 implies that $u'$ is not m-connecting

relative to $\mathbf{W}$ in $\mathcal{M}$. If a vertex other than $\{V_{i-1}, V_{i+1}\}$ blocks $u'$ in $\mathcal{M}$, the same vertex blocks $u$, which is a contradiction. Suppose without loss of generality that $V_{i-1}$ blocks $u'$ in $\mathcal{M}$. If $V_{i-1}$ is a collider on both $u$ and $u'$ or a non-collider on both of them, $V_{i-1}$ blocks $u$ in $\mathcal{M}$ which is a contradiction. So suppose $V_{i-1}$ is a collider on one of $u$ and $u'$, and a non-collider on the other one. From Lemma 7, $V_{i-1}, X \in Pa(V_{i+1})$. Also, $V_{i-1}$ is a collider on $u$ in this case, that is, $(V_{i-2}, V_{i-1}, X)$ is a collider path. From the assumption of the theorem, $V_{i-2} \in Adj(V_{i+1})$. The edge between $V_{i-2}$ and $V_{i+1}$ has an arrowhead at $V_{i+1}$, as otherwise an (almost) directed cycle is formed over $V_{i-2}, V_{i-1}, V_{i+1}$. Now define the path $u''$ as $u'' = (Y, V1, ..., V_{i-2}, V_{i+1}, ..., V_m, Z)$. This path also exists in $\mathcal{H}$, and therefore, $u''$ is not m-connecting relative to $\mathbf{W}$ in $\mathcal{H}$. Hence, $u''$ is not m-connecting relative to $\mathbf{W}$ in $\mathcal{M}$. If a vertex other than $V_{i-2}$ blocks $u''$ in $\mathcal{M}$, it also blocks $u'$ in $\mathcal{M}$, which is a contradiction, since we assumed that only $V_{i-1}$ blocks this path. If $V_{i-2}$ is a collider on both $u'$ and $u''$, or a non-collider on both of them, $V_{i-2}$ blocks $u'$ in $\mathcal{M}$, which is a contradiction. Now applying Lemma 7 implies that $V_{i-2} \in Pa(V_{i+1})$ and $(V_{i-3}, V_{i-2}, V_{i-1}, X)$ is collider path. Continuing in this manner finally implies that $Y \in Adj(V_{i+1})$ and the edge between $Y$ and $V_{i+1}$ has an arrowhead at $V_{i+1}$. Now since the path $(Y, V_{i+1}, ..., V_m, Z)$ is not m-connecting relative to $\mathbf{W}$, there exists a vertex $T$ that blocks it in $\mathcal{M}$. The same vertex must block $(Y, V_1, V_{i+1}, ..., V_m, Z)$, which is a contradiction. Note that now $T$ is either a collider on both of these paths, or a non-collider on both of them. Also note that the assumption that $V_{i-1}$ blocks $u'$ in $\mathcal{M}$ does not violate the generality of the proof as if we assumed that $V_{i+1}$ blocks $u'$, that would imply the same arguments for the paths $(Y, V_1, ..., V_{i-1}, V_j, V_{j+1}, ..., V_m, Z)$, with the only difference that $Y$ and $Z$ would be interchanged throughout the proof.

3. $X$ is a collider on $u$: Suppose $u = (Y, V_1, \ldots, V_{i-1}, V_i = X, V_{i+1}, \ldots, V_m, Z)$. If a vertex other than $X$ blocks $u$ in $\mathcal{M}$, we are done. Otherwise, we claim that $X$ blocks $u$ in $\mathcal{M}$. Since $X \notin \mathbf{W}$, it suffices to show that $De_{\mathcal{M}}(X) \cap (\{Y, Z\} \cup \mathbf{W}) = \varnothing$. Assume by contradiction that there exists a directed path from $X$ to a vertex in $\{Y, Z\} \cup \mathbf{W}$, and let $T \in Ch(X)$ denote the first vertex next to $X$ on this path. Note that $T \notin \{V_{i-1}, V_{i+1}\}$. Since $(V_{i-1}, X)$ and $(V_{i+1}, X)$ are collider paths and $X \in Pa(T)$, $V_{i-1}, V_{i+1} \in Adj(T)$ from the assumption. Both of these edges must have arrows on the side of $T$, as otherwise, an (almost) directed cycle would occur. Therefore, $T$ is a collider on $(V_{i-1}, T, V_{i+1})$. Now, consider the path $u' = (Y, V1, ..., V_{i-1}, T, V_{i+1}, ..., V_m, Z)$, which is a path in $\mathcal{H}$ and must not be m-connecting relative to $\mathbf{W}$ in $\mathcal{H}$. Hence, Lemma 6 implies that $u'$ is not m-connecting relative to $\mathbf{W}$ in $\mathcal{M}$. If a vertex other than $\{V_{i-1}, T, V_{i+1}\}$ blocks $u'$ in $\mathcal{M}$, the same vertex blocks $u$, which is a contradiction. $T$ cannot block $u'$ in $\mathcal{M}$ as it is a collider on $u'$ and it has a descendant in $\{Y, Z\} \cup \mathbf{W}$. Thus, suppose without loss of generality that $V_{i-1}$ blocks $u'$ in $\mathcal{M}$. If $V_{i-1}$ is a collider on both $u$ and $u'$ or a non-collider on both of them, $V_{i-1}$ blocks $u$ in $\mathcal{M}$ which is a contradiction. So suppose $V_{i-1}$ is a non-collider on $u'$ and a collider on $u$. Note that the other case is not possible because an (almost) directed cycle would occur over the vertices $V_{i-1}, X, T$. As a result, $V_{i-1} \in Pa(T)$. Now, consider the collider path $(V_{i-2}, V_{i-1}, X)$ in which $V_{i-1}, X \in Pa(T)$. Therefore, $V_{i-2} \in Adj(T)$. Again, this edge must have an arrowhead on the side of $T$, as otherwise an (almost) directed cycle is formed over $(V_{i-2}, V_{i-1}, T$. Now, consider the path $u'' = (Y, V1, \ldots, V_{i-2}, T, V_{i+1}, \ldots, V_m, Z)$, which is a path in $\mathcal{H}$, and therefore, is not m-connecting relative to $\mathbf{W}$ in $\mathcal{H}$. In this case, Lemma 6 implies that $u''$ is not m-connecting relative to $\mathbf{W}$ in $\mathcal{M}$. We can repeat the arguments above for this path, implying that either there exists a vertex that blocks $u$ in $\mathcal{M}$, or $V_{i-2} \in Pa(T)$, and therefore, $V_{i-3} \in Adj(T)$ (or alternatively, $V_{i+1} \in Pa(T)$, and therefore, $V_{i+2} \in Adj(T)$, which does not alter the proof.) Continuing in the same manner, either there exists a vertex that blocks $u$ in $\mathcal{M}$ which is a contradiction, or $Y, Z \in Adj(T)$, where $T$ is a collider on $(Z, T, Y)$. Finally, $(Z, T, Y)$ is a path in $\mathcal{H}$ and must not be m-connecting relative to $\mathbf{W}$, but this is not possible because $De_{\mathcal{M}}(Y) \cap Anc(\{Y, Z\} \cup \mathbf{W}) \neq \varnothing$. This contradiction proves that $X$ cannot have a descendant in $\{Y, Z\} \cup \mathbf{W}$, which implies that $X$ blocks $u$ in $\mathcal{M}$.

In all of the cases, $u$ is not m-connecting relative to $\mathbf{W}$, which completes the proof. $\square$

### A.2 Reduction to DAGs

The notion of removability is first discussed in [11] for the case of DAGs. Herein, we discuss how our definition of removability for MAGs (Definition 4) and the provided graphical representation (Theorem 1) can be reduced to their results when we restrict ourselves to the space of DAGs. Note

that our removability tests in Theorem 2 do not reduce to what they proposed for DAGs. For instance, we directly test the removability of a vertex without identifying its so-called co-parents.

- Definition 4: In the case of DAGs, m-separation reduces to d-separation. Hence, Definition 4 is reduced to what [11] proposed in the case of DAGs.

- Graphical representation: Suppose the ground-truth graph is a DAG. Note that collider paths in DAGs can be of length at most two and the vertices have no neighbors. In this case, our graphical representation of a removable variable in Theorem 1 is reduced to what is proposed in Theorem 5 of [11].

The removability test provided in [11] fails in the case that causal sufficiency is violated. Consider for example the vertex $X$ in Figure 2a. If the proposed tests of [11] are performed for $X$, then $Z$ and $V_1$ are identified to be adjacent to $X$, and then the collider paths $X \to Z \leftarrow V_1$, $X \to Z \leftarrow V_2$, and $X \to V_1 \leftarrow V_2$ are identified. Then due to their removability tests, $X$ is decided to be removable since the pairs $(Z, V_1)$, $(Z, V_2)$ and $(V_1, V_2)$ cannot be m-separated. However, we know from Theorem 1 that $X$ is not removable in this MAG.

# B Proofs

In this section, we first present fundamental lemmas used throughout our proofs. The proofs for the results of the main text is provided in Appendix B.2.

## B.1 Preliminary lemmas

**Lemma 1.** *Suppose $X$ is a vertex in a MAG $\mathcal{M}$ with vertex set $\mathbf{V}$ such that if $Y \in Pa(X)$ and $Z \in Ch(X)$, then $Y \in Pa(Z)$. Let $\mathcal{H}$ be the induced subgraph of $\mathcal{M}$ over $\mathbf{V} \setminus \{X\}$. Note that $\mathcal{H}$ is also a MAG. In this case, for any $Y \in \mathbf{V} \setminus \{X\}$,*

$$De_{\mathcal{M}}(Y) \setminus \{X\} = De_{\mathcal{H}}(Y).$$

*Proof.* Suppose $Z \in De_{\mathcal{M}}(Y) \setminus \{X\}$, i.e., there exists a directed path from $Y$ to $Z \neq X$ in $\mathcal{M}$. If this path does not pass through $X$, the same path exists in $\mathcal{H}$, and $Z \in De_{\mathcal{H}}(Y)$. Otherwise, suppose this path is $(Y, U_1, \ldots, U_i, X, U_{i+1}, \ldots, Z)$. Since $U_i \in Pa(X)$ and $U_{i+1} \in Ch(X)$, $U_i \in Pa(U_{i+1})$. Hence, $(Y, U_1, \ldots, U_i, U_{i+1}, \ldots, Z)$ is a directed path in $\mathcal{H}$, and $Z \in De_{\mathcal{H}}(Y)$. This implies that

$$De_{\mathcal{M}}(Y) \setminus \{X\} \subseteq De_{\mathcal{H}}(Y).$$

Furthermore, if there exists a directed path from $Y$ to $Z$ in $\mathcal{H}$, the same path exists in $\mathcal{M}$, which implies that

$$De_{\mathcal{H}}(Y) \subseteq De_{\mathcal{M}}(Y) \setminus \{X\}.$$

This completes the proof. □

**Lemma 2.** *Let $X$ and $Y$ be two non-adjacent vertices in a MAG $\mathcal{M}$, where $X \notin Anc(Y)$. Then*

$$(X \perp Y | \mathbf{W} \setminus \{X, Y\})_{\mathcal{M}}, \quad where \ \mathbf{W} = N(X) \cup \big(Pa^+(X) \cap Anc(\{X, Y\})\big). \tag{9}$$

*Proof.* Let $u = (X = V_0, V_1, \ldots, V_m, Y = V_{m+1})$ be an arbitrary path between $X$ and $Y$. It suffices to show that $\mathbf{W} \setminus \{X, Y\}$ blocks $u$. Let $i$ be the largest index such that all the edges on $(V_0, V_1, ..., V_i)$ are bidirectional. We consider the following cases separately.

1. $i \geq m$: In this case, all the vertices $V_1, ..., V_m$ on the path are colliders that belong to $Pa^+(X)$. Since $X$ and $Y$ are non-adjacent, $u$ is not an inducing path. Hence, there exists $j$ such that $V_j \notin Anc(\{X, Y\})$ and therefore, $V_j \notin Anc(\mathbf{W} \cup \{X, Y\})$. Hence, $V_j$ blocks $u$.

2. $i = 0$: If $V_1 \in Pa(X) \cup N(X)$, then $V_1 \in \mathbf{W} \setminus \{X, Y\}$ is a non-collider on $u$ that blocks $u$. Otherwise, $V_1 \in Ch(X)$. Continuing the path $u$ from $V_1$, let $V_j$ be the first collider on $u$. Note that such a collider exists as $X \notin Anc(Y)$ and therefore, $u$ is not a directed path. $V_j$ is a descendant of $X$ and therefore, $V_j \notin Anc(X, Y)$. Hence, $V_j \notin Anc(\mathbf{W} \cup \{X, Y\})$ blocks $u$ as a collider.

3. $1 \leq i < m$: The edge between $V_i$ and $V_{i+1}$ is either $V_i \to V_{i+1}$, or $V_i \leftarrow V_{i+1}$ (it cannot be undirected by definition of MAGs.) Let $Z$ be the parent among these two vertices, and $T$ be the child, i.e., if $V_i \to V_{i+1}$, then $Z$ and $T$ denote $V_i$ and $V_{i+1}$, respectively. Note that $Z \in Pa^+(X)$. If $Z \in Anc(\{X,Y\})$, then $Z \in \mathbf{W} \setminus \{X,Y\}$ blocks $u$ as a non-collider. Suppose otherwise that $Z \notin Anc(\{X,Y\})$. Continuing the path $u$ from $Z$ towards the side of $T$, let $V_j$ be the first collider. Such a collider exists as $Z \notin Anc(\{X,Y\})$. $V_j$ is a descendant of $Z$, and therefore $V_j \notin Anc(\{X,Y\})$. Hence, $V_j \notin Anc(\mathbf{W} \cup \{X,Y\})$ blocks $u$ as a collider.

In all of the above cases, $\mathbf{W} \setminus \{X,Y\}$ blocks $u$, which completes the proof. $\qquad \square$

**Lemma 3.** *If $X \in \mathbf{V}$ is a removable vertex, then for any $Y, Z \in Mb(X)$,*

$$Z \in Mb(Y) \text{ and } Y \in Mb(Z).$$

*Moreover, there exists at least one collider path between $Y$ and $Z$ that passes through only the vertices in $Mb(X) \cup \{X\}$.*

*Proof.* Take two arbitrary vertices $Y, Z$ in $Mb(X)$. We will show that there exists a collider path between $Y$ and $Z$ that passes through only the vertices in $Mb(X) \cup \{X\}$.

Since $Y, Z \in Mb(X)$, there exist collider paths $(Y, V_1, \ldots, V_i, X)$ and $(X, W_1, \ldots, W_j, Z)$, where $V_1, ..., V_i, W_1, ..., W_j \in Mb(X)$. Consider the path $(Y, V_1, \ldots, V_i, X, W_1, \ldots, W_j, Z)$. If $X$ is a collider on this path, we are done. Otherwise, without loss of generality, assume $W_1 \in Ch(X) \cup N(X)$. Since $X$ is removable, $V_i \in Adj(W_1)$. We now consider the following two cases separately.

1. $W_1 \in Ch(X)$: If the edge between $V_i$ and $W_1$ is bidirected, then the path $(Y, V_1, \ldots, V_i, W_1, \ldots, W_j, Z)$ is a collider path. Otherwise, again without loss of generality assume $V_i$ is a parent of $W_1$. Note that a child of $X$ cannot be a parent of its spouse since this would create an almost directed cycle. Now, since $X$ is removable, $V_{i-1}$ and $W_1$ are adjacent. If the edge is bidirected, then $(Y, V_1, \ldots, V_{i-1}, W_1, \ldots, W_j, Z)$ is a collider path. Otherwise, we can continue the same argument as before by induction on $i$ and conclude that $Y$ is adjacent to $W_1$. Since the structure is a MAG, $W_1 \notin Pa(Y)$ and $W_1$ is a collider on $(Y, W_1, \ldots, W_j, Z)$. Therefore, a collider path exists between $Y$ and $Z$ using only the vertices in $Mb(X) \cup \{X\}$.

2. $W_1 \in N(X)$: In this case, $W_1 = Z$, since $W_1$ is not a collider. Also, since $X$ has a neighbor, it cannot have a parent or a spouse. As a result, $V_i \in Ch(X) \cup N(X)$. If $V_i \in N(X)$, then by the same argument, $V_i = Y$ and we already know that $Y$ and $Z$ are adjacent, which is the desired collider path. Otherwise, $Z \in Pa(V_i)$. Now, the path $(Y, V_1, \ldots, V_i, Z)$ is the desired path, which completes the proof.

$\qquad \square$

**Lemma 4.** *Suppose $\mathbf{V} \subseteq \mathcal{O}$ and let $\mathcal{G} = \mathcal{G}_{\boldsymbol{\mathcal{V}}}[\mathbf{V}|\boldsymbol{\mathcal{S}}]$. If $X \in \mathbf{V}$ is removable in $\mathcal{G}$, then for any $Y, Z \in \mathbf{V} \setminus \{X\}$ and $\mathbf{W} \subseteq \mathbf{V} \setminus \{X,Y,Z\}$,*

$$(Y \perp Z | \mathbf{W} \cup \{X\})_{\mathcal{G}} \implies (Y \perp Z | \mathbf{W})_{\mathcal{G}}.$$

*Proof.* Suppose $(Y \perp Z | \mathbf{W} \cup \{X\})_{\mathcal{G}}$. We need to show that $(Y \perp Z | \mathbf{W})_{\mathcal{G}}$. To this end, we first show that $(Y \perp Z | \mathbf{W})_{\mathcal{H}}$, where $\mathcal{H}$ is the induced subgraph of $\mathcal{G}$ over $\mathbf{V} \setminus \{X\}$.

Note that all the paths between $Y$ and $Z$ are blocked by $\mathbf{W} \cup \{X\}$ in $\mathcal{G}$. Now, take an arbitrary path $u$ between $Y$ and $Z$ in $\mathcal{H}$. This path also exists in $\mathcal{G}$, and $X$ is not on the path. We claim $\mathbf{W}$ blocks it in $\mathcal{H}$. Suppose $u$ is blocked by a vertex $T$ in $\mathcal{G}$ (note that $T \neq X$.) If $T$ is a non-collider on $u$, then it also blocks $u$ in $\mathcal{H}$. If it is a collider with no descendants in $\mathbf{W} \cup \{X\}$, then lemma 1 implies that $De_{\mathcal{H}}(T) \cap \mathbf{W} = \varnothing$, and $T$ blocks $u$ in $\mathcal{H}$. Therefore, $(Y \perp Z | \mathbf{W})_{\mathcal{H}}$.

Finally, since $X$ is removable in $\mathcal{G}$ and $(Y \perp Z | \mathbf{W})_{\mathcal{H}}$, Definition 4 implies that $(Y \perp Z | \mathbf{W})_{\mathcal{G}}$. $\quad \square$

**Lemma 5.** *Suppose $(X, V_1, ..., V_m, Y)$ is a collider path where $\{X, V_1, ..., V_m\} \in Pa(Z)$ for a vertex $Z$. If $(Y \perp Z | \mathbf{W})$ for a set $\mathbf{W}$, then $X \in \mathbf{W}$.*

*Proof.* Since $Y$ and $Z$ are m-separated by $\mathbf{W}$, $\mathbf{W}$ blocks all the paths between $Y$ and $Z$. Now consider the path $u = (Z, X, V_1, ..., V_m, Y)$ which must be blocked by $\mathbf{W}$. $\{V_1, ..., V_m\} \subseteq Anc(Z)$ are colliders on $u$. As a result, if $X \notin \mathbf{W}$, then $u$ is m-connecting relative to $\mathbf{W}$, which is a contradiction. $\qquad\square$

**Lemma 6.** *Suppose $\mathcal{G}$ is a MAG with the vertex set $\mathbf{V}$, and $X \in \mathbf{V}$ is removable in $\mathcal{G}$. Let $\mathcal{H}$ denote the induced subgraph of $\mathcal{G}$ over $\mathbf{V} \setminus \{X\}$. For a path $u$ in $\mathcal{H}$ and a set $\mathbf{W} \subseteq \mathbf{V} \setminus \{X\}$,*

$$u \text{ is m-connecting w.r.t. } \mathbf{W} \text{ in } \mathcal{M} \iff u \text{ is m-connecting w.r.t. } \mathbf{W} \text{ in } \mathcal{H}. \qquad (10)$$

*Proof.* The proof of both sides of Equation (10) are the same. Let $\mathcal{G}_1$ be $\mathcal{M}$ or $\mathcal{H}$, and $\mathcal{G}_2$ be the other one. Suppose $\mathbf{W} \subseteq \mathbf{V} \setminus \{X\}$ and let $u = (Y, V_1, \ldots, V_m, Z)$ be a path in $\mathcal{H}$ such that $u$ is m-connecting relative to $\mathbf{W}$ in $\mathcal{G}_1$. We need to show that $u$ is m-connecting relative to $\mathbf{W}$ in $\mathcal{G}_2$. Let $T$ be an arbitrary non-endpoint vertex on $u$. We need to show that $T$ does not block $u$ in $\mathcal{G}_2$. There are two possibilities.

1. $T$ is non-collider in $u$: Since $T$ does not block $u$ in $\mathcal{G}_1$, $T \notin \mathbf{W}$. Hence, $T$ does not block $u$ in $\mathcal{G}_2$.

2. $T$ is a collider on $u$: Since $T$ does not block $u$ in $\mathcal{G}_1$, $De_{\mathcal{G}_1}(T) \cap (\mathbf{W} \cup \{Y, Z\}) \neq \varnothing$. Hence, Lemma 1 implies that $De_{\mathcal{G}_2}(T) \cap (\mathbf{W} \cup \{Y, Z\}) \neq \varnothing$ and $T$ does not block $u$ in $\mathcal{G}_2$.

In both cases $T$ does not block $u$ in $\mathcal{G}_2$ and therefore, $u$ is m-connecting relative to $\mathbf{W}$ in $\mathcal{G}_2$. $\qquad\square$

**Lemma 7.** *Suppose $\mathcal{G}$ is a MAG and $u = (Y, ..., V_0, V_1, X, V_2, ..., Z)$ is a path in $\mathcal{G}$, where $X$ is a non-collider on $u$ and $V_1 \in Adj(V_2)$. Define $\tilde{u} = (Y, ..., V_1, V_2, ..., Z)$, which is a path in $\mathcal{G}$. If $V_1$ is a collider on $u$ and a non-collider on $\tilde{u}$, or a non-collider on $u$ and a collider on $\tilde{u}$, then $X, V_1 \in Pa(V_2)$.*

*Proof.* First note that the edge between $V_0$ and $V_1$ must have an arrowhead at $V_1$, since otherwise $V_1$ cannot be a collider on any of the paths. Now, two possibilities may occur.

- The edge between $V_1$ and $X$ has a tail at $V_1$: Since $V_1$ has an arrowhead, it does not have any neighbors, i.e., $X \notin N(V_1)$. Hence, $V_1 \in Pa(X)$. As $X$ is not a collider on $u$, $X \in Pa(V_2)$, i.e., $V_1 \to X \to V_2$. Now, the edge between $V_1$ and $V_2$ can only be $V_1 \to V_2$, as otherwise, an (almost) directed cycle is formed on $V_1, X, V_2$.

- The edge between $V_1$ and $X$ has an arrowhead at $V_1$: Since $V_1$ is a collider on $u$, it is a non-collider on $\tilde{u}$. Also, $V_1$ does not have any neighbors by definition of MAGs, which implies that $V_1 \in Pa(V_2)$. Consider the edge between $X$ and $V_2$. If this edge has an arrowhead at $X$, then $X \in Pa(V_1)$ as $X$ is a non-collider on $u$. Now, the triple $X, V_1, V_2$ forms an (almost) directed cycle, which is a contradiction. As a result, the edge between $X$ and $V_2$ has a tail at $X$. Note that $V_2$ has no neighbors because $V_1 \to V_2$. This implies that $X \in Pa(V_2)$, which competes the proof.

$\qquad\square$

## B.2 Main Results

**Proposition 1.** *Suppose $\mathbf{V} \subseteq \mathcal{O}$ and $X \in \mathbf{V}$. $\mathcal{G}_{\mathcal{V}}[\mathbf{V} \setminus \{X\} | \mathcal{S}]$ is equal to the induced subgraph of $\mathcal{G}_{\mathcal{V}}[\mathbf{V} | \mathcal{S}]$ over $\mathbf{V} \setminus \{X\}$ if and only if $X$ is removable in $\mathcal{G}_{\mathcal{V}}[\mathbf{V} | \mathcal{S}]$.*

*Proof.* Denote $\mathcal{G}_{\mathcal{V}}[\mathbf{V} | \mathcal{S}]$, $\mathcal{G}_{\mathcal{V}}[\mathbf{V} \setminus \{X\} | \mathcal{S}]$ and the induced subgraph of $\mathcal{G}_{\mathcal{V}}[\mathbf{V} | \mathcal{S}]$ over $\mathbf{V} \setminus \{X\}$ by $\mathcal{G}$, $\mathcal{M}$ and $\mathcal{H}$, respectively.

**only if:** Suppose $\mathcal{M}$ is equal to $\mathcal{H}$. Let $Y$ and $W$ be arbitrary vertices in $\mathbf{V} \setminus \{X\}$ and $\mathbf{Z}$ be an arbitrary subset of $\mathbf{V} \setminus \{X\}$. It suffices to show that Equation (4) holds. Since m-separation and conditional independence are equivalent in latent projections $\mathcal{G}$ and $\mathcal{M}$,

$$(Y \perp W | \mathbf{Z})_{\mathcal{G}} \Leftrightarrow (Y \perp\!\!\!\perp W | \mathbf{Z}) \Leftrightarrow (Y \perp W | \mathbf{Z})_{\mathcal{M}} \Leftrightarrow (Y \perp W | \mathbf{Z})_{\mathcal{H}},$$

where the last equivalence is due to the fact that $\mathcal{M}$ and $\mathcal{H}$ are equal.

**if:** Suppose $X$ is removable. We first prove that the skeleton of $\mathcal{M}$ and $\mathcal{H}$ are equal. With similar arguments to the above case, CI relations and m-separation in $\mathcal{G}$ and $\mathcal{M}$ are equivalent. Therefore,

$$(Y \perp W|\mathbf{Z})_{\mathcal{M}} \Leftrightarrow (Y \perp\!\!\!\perp W|\mathbf{Z}) \Leftrightarrow (Y \perp W|\mathbf{Z})_{\mathcal{G}} \Leftrightarrow (Y \perp W|\mathbf{Z})_{\mathcal{H}},$$

where the last equivalence follows from Equation (4). Since $\mathcal{M}$ and $\mathcal{H}$ impose the same set of m-separations, that is they are Markov equivalent, they must have the same skeleton. Now for the edge marks, note that the edge marks of $\mathcal{H}$ are those of $\mathcal{G}$, as $\mathcal{H}$ is an induced subgraph of $\mathcal{G}$. Furthermore, edges in $\mathcal{G}$ and $\mathcal{M}$ are oriented by the same rules of Definition 2 as they are the projections of the same DAG $\mathcal{G}_{\boldsymbol{\mathcal{V}}}$. Therefore, both the skeleton and the edge marks of $\mathcal{M}$ and $\mathcal{H}$ are identical, which completes the proof. $\qquad\square$

**Theorem 2.** *Suppose the edge-induced subgraph of $\mathcal{M}$ over the undirected edges (i.e., the edges due to selection bias) is chordal. Let $\mathcal{G} = \mathcal{G}_{\boldsymbol{\mathcal{V}}}[\mathbf{V}|\boldsymbol{\mathcal{S}}]$ for some $\mathbf{V} \subseteq \mathcal{O}$. $X \in \mathbf{V}$ is removable in $\mathcal{G}$ if and only if for every $Y \in Adj(X)$ and $Z \in Mb_{\mathbf{V}}(X)$, at least one of the following holds.*

       ***Condition 1:*** $\exists \mathbf{W} \subseteq Mb_{\mathbf{V}}(X)\backslash\{Y, Z\}\colon\ Y \perp\!\!\!\perp Z|\mathbf{W}.$

       ***Condition 2:*** $\forall \mathbf{W} \subseteq Mb_{\mathbf{V}}(X)\backslash\{Y, Z\}\colon\ Y \not\!\perp\!\!\!\perp Z|\mathbf{W} \cup \{X\}.$

*Furthermore, the set of removable vertices in $\mathcal{G}$ is non-empty.*

*Proof.* We first prove the equivalence of removability and the two conditions.

**only if:** Suppose $X$ is removable. It suffices to show that if Condition 2 does not hold, then condition 1 holds. Let $\mathbf{W_1} \subseteq Mb_{\mathbf{V}}(X)\backslash\{Y, Z\}$ be such that $Y \perp\!\!\!\perp Z|\mathbf{W_1} \cup \{X\}$. Since m-separation is equivalent to conditional independence, $(Y \perp Z|\mathbf{W_1}\cup\{X\})_{\mathcal{G}}$. Now from lemma 4, $(Y \perp Z|\mathbf{W_1})_{\mathcal{G}}$, which implies $(Y \perp\!\!\!\perp Z|\mathbf{W_1})$, that is, Condition 1 holds.

**if:** We show that the graphical representation of Theorem 1 is satisfied. To this end, we show $Y$ and $Z$ are adjacent in all of the following cases:

1. $u = (X, V_1, ..., V_m, Y)$ is a collider path such that $\{X, V_1, ..., V_m\} \subseteq Pa(Z)$: By definition of $Pa^+(\cdot)$, $Pa^+(Z) \subseteq Mb_{\mathbf{V}}(X) \cup \{X\}$. Lemma 2 indicates that

   $$\mathbf{W_1} = (Pa^+(Z) \cap Anc(\{Z, Y\}))\backslash\{Z, Y\} \subseteq Mb_{\mathbf{V}}(X) \cup \{X\}$$

   m-separates $Y$ and $Z$. Note that $N(Z) = \varnothing$ since $Z$ has at least one parent. Since conditional independence is equivalent to m-separation,

   $$(Y \perp\!\!\!\perp Z|\mathbf{W_1}),$$

   that is, Condition 2 does not hold. If $Y$ and $Z$ are m-separated by some set $\mathbf{W_1}$, from Lemma 5, $X \in \mathbf{W_1}$. As a result, Condition 1 cannot hold for any $\mathbf{W} \subseteq Mb_{\mathbf{V}}(X)$ as these sets do not contain $X$, which is a contradiction. This proves that $Y$ and $Z$ are adjacent.

2. $Y \in Adj(X)$ and $Z \in Ch(X)$: The proof in this case is exactly the same as the previous one.

3. $Z \in N(X)$ and $Y \in Adj(X)$: Since $X$ has a neighbor, by definition of MAG, $Y$ is either a child or a neighbor of $X$. If $Y \in Ch(X)$, this case reduces to case 2 with $Y$ and $Z$ interchanged. So we only consider the case where $Y \in N(X)$. Considering the path $(Y, X, Z)$, no set $\mathbf{W}$ can m-separate $Y$ and $Z$ if $X \notin \mathbf{W}$, i.e., Condition 1 does not hold. We claim if $Y$ and $Z$ are not adjacent, Condition 2 does not hold either, which is a contradiction. To prove this, take $\mathbf{W} = \{X\} \cup N(X)\backslash\{Y, Z\}$. It is enough to show that $(Y \perp Z|\mathbf{W})_{\mathcal{G}}$, i.e., $\mathbf{W}$ blocks all the paths between $Y$ and $Z$. Let $u$ be an arbitrary path of length at least 2 between $Y$ and $Z$. If $u$ contains a directed or bidirected edge, it also contains a collider, since $Y$ and $Z$ do not have any incoming edges incident to them and therefore no ancestors. This collider blocks the path as it does not have any descendants in $\mathbf{W}$ (note that the vertices in $\mathbf{W}$ have at least one neighbor, and therefore by definition of MAG, they do not have any ancestors.) Otherwise, $u$ is a path with only undirected edges. If $X$ is on $u$, $X$ itself blocks this path. Otherwise, consider the cycle formed by adding the path $Y - X - Z$ to $u$. Since the edge-induced subgraph of $\mathcal{M} = \mathcal{G}_{\boldsymbol{\mathcal{V}}}[\mathcal{O}|\boldsymbol{\mathcal{S}}]$ over its undirected edges is chordal, if $Y$ and $Z$ are not adjacent, there exists a chord which connects $X$ to a non-endpoint vertex on $u$. As a result, at least one of the neighbors of $X$ appears on $u$, and therefore blocks $u$ as a non-collider, as it belongs to $\mathbf{W}$.

For a proof of the second part of the theorem, i.e., the set of removable vertices is non-empty, we refer the reader to Lemma 9 in Appendix D. □

**Theorem 3.** *Suppose the distribution $P_{\mathcal{V}}$ over $\mathcal{V} = \mathcal{O} \cup \mathcal{L} \cup \mathcal{S}$ is faithful to the DAG $\mathcal{G}_{\mathcal{V}}$. If the conditional independence relations among all variables in $\mathcal{O}$ given $\mathcal{S}$ is provided to L-MARVEL, the output of L-MARVEL is the PAG representing the Markov equivalence class of $\mathcal{G}_{\mathcal{V}}[\mathcal{O}|\mathcal{S}]$.*

*Proof.* In order to prove this theorem, it is enough to show that the information stored in $\mathcal{A}$, i.e., the set of adjacencies and the separating sets for non-adjacent variables, is correct. L-MARVEL identifies that two variables are not adjacent, only if it finds a separating set for them. In this case, L-MARVEL adds that separating set to $\mathcal{A}$. Hence, all the separating sets found in $\mathcal{A}$ are correct, and the non-adjacent variables in $\mathcal{A}$ are non-adjacent in $\mathcal{M}$. Note that even in the case that two variables are excluded from each other's Markov boundary, this is due to a found separating set for these two variables. It is left to show that L-MARVEL correctly finds all the adjacent variables in $\mathcal{M}$.

Let $\mathcal{H}_{\mathbf{V}}$ denote the induced subgraph of $\mathcal{M}$ over $\mathbf{V} \subseteq \mathcal{O}$. We claim every time that L-MARVEL is called over a subset $\mathbf{V} \subseteq \mathcal{O}$ during the execution of the algorithm, $\mathcal{H}_{\mathbf{V}}$ is equal to $\mathcal{G}_{\mathcal{V}}[\mathbf{V}|\mathcal{S}]$. For the first time, we call L-MARVEL over $\mathcal{O}$ and the claim holds. Now, assume $\mathcal{H}_{\mathbf{V}} = \mathcal{G}_{\mathcal{V}}[\mathbf{V}|\mathcal{S}]$ in a recursion. We need to show that our claim holds for the next recursion. First, note that Equation (2) implies that $\mathcal{H}_{\mathbf{V}}$ satisfies faithfulness with respect to $P_{\mathbf{V}|\mathcal{S}}$. Theorem 2 implies that when the if condition in line 9 holds for the first $i = i^*$, then $X_{i^*}$ is removable in $\mathcal{H}_{\mathbf{V}}$. Note that by Lemma 9, there always exists a variable that satisfies the if condition in line 9. Hence, Proposition 1 implies that in the next recursion, $\mathcal{H}_{\mathbf{V} \setminus X_{i^*}} = \mathcal{G}_{\mathcal{V}}[\mathbf{V} \setminus X_{i^*}|\mathcal{S}]$, which proves our claim.

So far we have shown that in each recursion, $\mathcal{H}_{\mathbf{V}} = \mathcal{G}_{\mathcal{V}}[\mathbf{V}|\mathcal{S}]$ and $\mathcal{H}_{\mathbf{V}}$ satisfies faithfulness with respect to $P_{\mathbf{V}|\mathcal{S}}$. Hence, Function **FindAdjacent** and **UpdateMb** correctly learn the adjacent variables and update the Markov boundaries, respectively. Hence, L-MARVEL manages to terminate after $n$ recursion and correctly add all the edges of $\mathcal{M}$ to $\mathcal{A}$. □

**Proposition 2.** *If $X$ is a removable variable in MAG $\mathcal{H}$ with vertices $\mathbf{V}$, then $|Mb_{\mathbf{V}}(X)| \leq \Delta_{in}^{+}(\mathcal{H})$.*

*Proof.* Consider the set of variables $Mb(X) \cup \{X\}$. Since MAGs are acyclic, there exists a vertex in this set such that it has no children in $Mb(X) \cup \{X\}$. Denote this vertex by $Z$. From Lemma 3, every vertex in $\{X\} \cup Mb(X) \setminus \{Z\}$ has a collider path to $Z$ such that it passes through only the vertices in $\{X\} \cup Mb(X)$. Since $Z$ has no child in this set, the vertex adjacent to $Z$ on these collider paths is either a parent, or a spouse, or a neighbor of $Z$. Therefore, by definition,

$$\{X\} \cup Mb(X) \setminus \{Z\} \subseteq Pa^{+}(Z).$$

As a result,

$$|Mb(X)| = |\{X\} \cup Mb(X) \setminus \{Z\}| \leq |Pa^{+}(Z)| \leq \Delta_{in}^{+}(\mathcal{H}).$$

□

**Proposition 3.** *The number of conditional independence tests Algorithm 1 performs on a MAG $\mathcal{M}$ of order $n$, in the worst case, is upper bounded by*

$$\mathcal{O}(n^2 + n\Delta_{in}^{+}(\mathcal{M})^2 2^{\Delta_{in}^{+}(\mathcal{M})}). \tag{11}$$

*Proof.* Algorithm 1 performs CI tests throughout the following subroutines:

- ComputeMb: This is the initial Markov boundary discovery, that can be performed using any of the existing quadratic algorithms such as GS, TC, IAMB, etc. as discussed in the main text, that is, $\mathcal{O}(n)$ CI tests are required for this task.

- **FindAdjacent($X$):** The performed CI tests are of the type $(X \perp\!\!\!\perp Y|\mathbf{W})$, where $Y \in Mb_{\mathbf{V}}(X)$ and $\mathbf{W} \subseteq Mb_{\mathbf{V}}(X) \setminus \{Y\}$. There are $|Mb_{\mathbf{V}}(X)|$ choices for $Y$ and $2^{(|Mb_{\mathbf{V}}(X)|-1)}$ choices for $\mathbf{W}$, that is, $|Mb_{\mathbf{V}}(X)| 2^{(|Mb_{\mathbf{V}}(X)|-1)}$ total tests.

- **IsRemovable($X$):** The performed CI tests are of the type $(Y \perp\!\!\!\perp Z|\mathbf{W})$, where $Y \in Adj(X) \cap \mathbf{V}$, $Z \in Mb_{\mathbf{V}}(X) \setminus \{Y\}$ and $\mathbf{W} \subseteq \{X\} \cup Mb_{\mathbf{V}}(X) \setminus \{Y, Z\}$. There are $|N(X)|$ choices for $Y$, at most $|Mb_{\mathbf{V}}(X)|$ choices for $Z$ and $2^{(|Mb_{\mathbf{V}}(X)|-1)}$ choices for $\mathbf{W}$, that is, at most $|Mb_{\mathbf{V}}(X)||N(X)| 2^{(|Mb_{\mathbf{V}}(X)|-1)}$ total tests.

- **UpdateMb($X$):** L-MARVEL performs a single CI test for any pair of vertices in $Mb_{\mathbf{V}}(X)$, that is $\binom{|Mb_{\mathbf{V}}(X)|}{2}$ tests.

Note that due to Proposition 2, the for loop in line 6 of Algorithm 1 only reaches vertices with maximum Markov boundary size of $\Delta_{\text{in}}^+(\mathcal{M})$. Therefore, the number of CI tests performed for a single vertex $X$ is upper bounded by $\mathcal{O}(\Delta_{\text{in}}^+(\mathcal{M})^2 2^{\Delta_{\text{in}}^+(\mathcal{M})})$. We shall next discuss why we do not need to perform each of the aforementioned tests more than once, which the yields the desired upper bound.

- **FindAdjacent($X$):** The set of vertices adjacent to $X$ does not change throughout the algorithm. Therefore, the first time that **FindAdjacent** is called for $X$, the variables adjacent to $X$ are identified and saved in $\mathcal{A}$, and are used in later iterations without requiring further CI tests.

- **IsRemovable($X$):** It might happen that L-MARVEL performs some CI tests to identify that $X$ is not removable, and therefore, it has to call **IsRemovable** for $X$ in a later iteration (note that every variable gets removed throughout the algorithm.) This is due to the fact that the removal of other variables can render $X$ removable in a later iteration. However, we claim that no duplicate CI tests are needed in later iterations where L-MARVEL calls **IsRemovable**. To show this, note that for any pair $Y, Z$ where $Y \in Adj(X) \cap \mathbf{V}$ and $Z \in Mb_{\mathbf{V}}(X) \setminus \{Y\}$, all of the separating sets of $Y$ and $Z$ in $Mb_{\mathbf{V}}(X) \cup \{X\}$ are saved in $\mathcal{A}$ during the first call to **IsRemovable**. Since the Markov boundary of $X$ can only be reduced throughout the algorithm, in all the succeeding iterations, it suffices for L-MARVEL to query the found separating sets.

- **UpdateMb($X$):** These CI tests are performed only before $X$ is removed from the set of variables, that is, they are performed exactly once for each variable.

$\square$

**Theorem 4.** *The number of conditional independence tests of the form $(X \perp\!\!\!\perp Y | \mathbf{Z})$ required by any constraint-based algorithm on a MAG $\mathcal{M}$ of order $n$, in the worst case, is lower bounded by*

$$\Omega(n^2 + n\Delta_{in}^+(\mathcal{M}) 2^{\Delta_{in}^+(\mathcal{M})}). \tag{12}$$

*Proof.* First, suppose an algorithm does not query any CI test of the form $(X \perp\!\!\!\perp Y | \mathbf{W})$ for a pair of vertices $(X, Y)$. If all the queried CI tests yield independence, this algorithm cannot tell an empty graph and a graph where only $X$ and $Y$ are adjacent apart. Therefore, at least one CI test is required for any pair of vertices, which yields a lower bound of $\binom{n}{2}$.

Furthermore, [11] proposed a lower bound of the form $\Omega(n\Delta_{in}(\mathcal{M}) 2^{\Delta_{in}(\mathcal{M})})$ for the case that $\mathcal{M}$ is a DAG, where $\Delta_{in}(\mathcal{M})$ is the maximum number of parents among the variables. Note that in the case of a DAG, $\Delta_{\text{in}}^+(\mathcal{M}) = \Delta_{in}(\mathcal{M})$, which proves our claim. However, we briefly discuss how their worst-case example can be modified in a way that it is no longer a DAG, and also $\Delta_{\text{in}}^+(\mathcal{M})$ is strictly larger than $\Delta_{in}(\mathcal{M})$. The provided example is as follows. The vertices of the ground truth graph is partitioned into $\frac{n}{\Delta_{\text{in}}^+(\mathcal{M})+1}$ clusters, where each cluster is a complete graph and there is no edge between the variables of different clusters. They show that if fewer CI tests than the claimed lower bound are performed, then a CI test of the form $(X \perp\!\!\!\perp Y | \mathbf{W_1} \cup \mathbf{W_2})$ is not queried, where $X, Y, \mathbf{W_1}$ belong to a cluster $\mathbf{C}$, whereas $\mathbf{W_2}$ does not contain any vertex of $\mathbf{C}$. Then they show that the graph where $\mathbf{W_1}$ are parents of $X$ and $Y$, and the rest of the graph is exactly the same as $\mathcal{M}$ with the exception that there is no edge between $X$ and $Y$ is consistent with the performed CI tests. In this example, if the rest of the edges in the cluster $\mathbf{C}$, i.e., the edges other than those between $\mathbf{W}$ and $X, Y$, as well as all the edges in the other clusters are replaced by bidirectional edges, the same proof still works. Note that in this example, $\Delta_{\text{in}}^+(\mathcal{M}) = |\mathbf{C}| - 1$, whereas $\Delta_{in} = |\mathbf{W_1}|$. Hence, we achieve the lower bound of Equation (12). $\square$

## C   Additional experiments

In this section, we provide further experimental results to assess the performance of L-MARVEL against the state of the art.

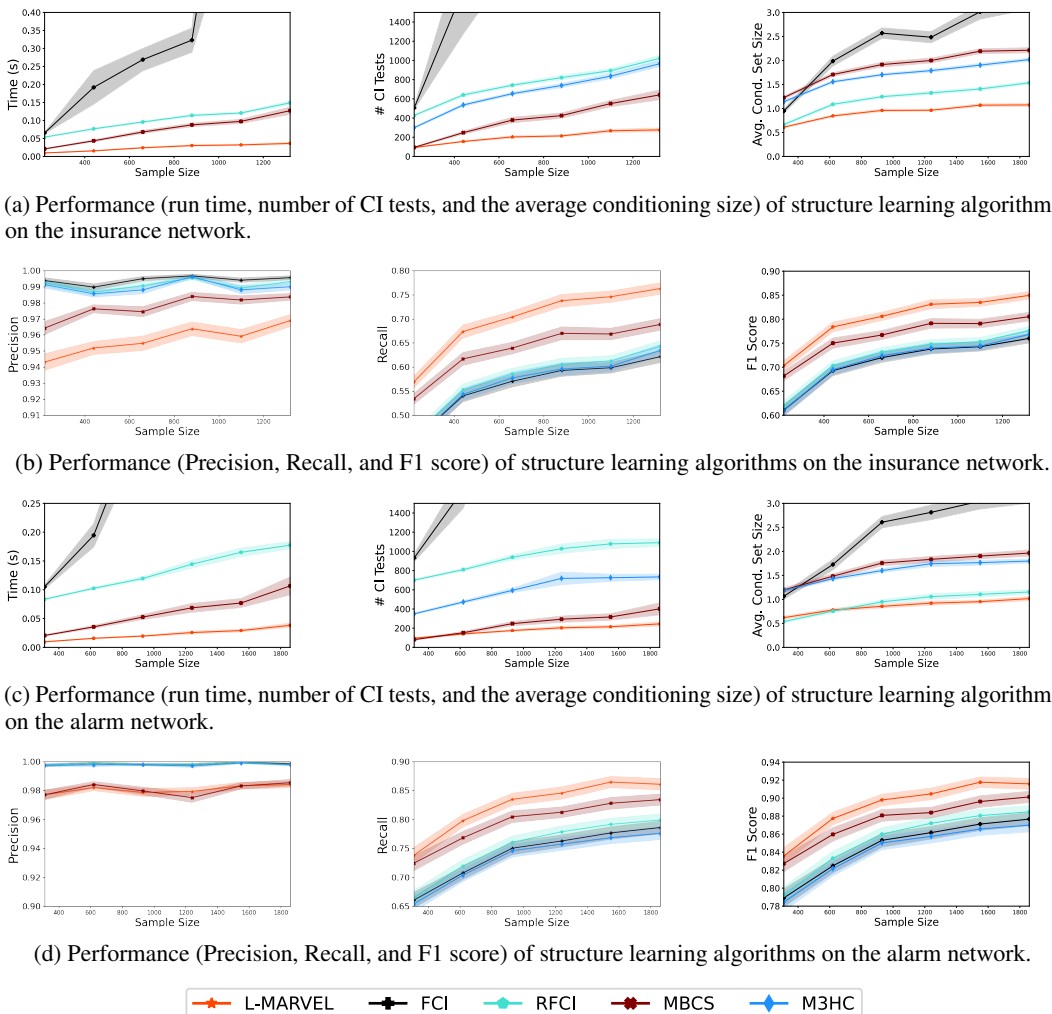

(a) Performance (run time, number of CI tests, and the average conditioning size) of structure learning algorithms on the insurance network.

(b) Performance (Precision, Recall, and F1 score) of structure learning algorithms on the insurance network.

(c) Performance (run time, number of CI tests, and the average conditioning size) of structure learning algorithms on the alarm network.

(d) Performance (Precision, Recall, and F1 score) of structure learning algorithms on the alarm network.

Figure 4: Effect of the sample size on the performance of structure learning algorithms on two benchmark structures, where the sample size varies from $= 10|\mathcal{O}|$ to $= 60|\mathcal{O}|$. The parameters of the experiments are preserved as in Table 1, except for the sample size.

Figure 4 illustrates the effect of the sample size on the performance of various algorithms. It is seen that L-MARVEL has the lowest run time and the fewest number of performed CI tests, while it maintains high accuracy in the wide range of the sample size. Also note that on these benchmark structures, L-MARVEL beats RFCI in terms of the average number of CI tests, which was the only metric in which RFCI showed advantage on random graphs. The experimental setting in this part is exactly that of Table 1, except for the sample size, to observe only the effect of the sample size. Each point of these graphs represents 50 MAGs generated by selecting the latent and selection variables uniformly at random.

Table 2 extends our experiments to two new benchmark structures, namely mildew and water. The number of latent and selection variables varies in different columns of this table, where the latent and selection variables are chosen uniformly at random. The coefficients of the linear SEM are chosen uniformly at random from the interval $\pm(1, 1.5)$, whereas the standard deviation of the noise variables is chosen uniformly at random from the interval $(1, \sqrt{2})$ to represent a set of parameters different than that of the main text. The entries of the table represent an average of 20 runs. As observed in Table 1, L-MARVEL outperforms all the other algorithms in almost every comparison metric, except for the precision, where it still is competent to the state of the art.

Table 2: Performance of various algorithms on the benchmark structures, when sample size $= 50|\mathcal{O}|$.

| | Structure $(|\mathcal{O}|,|\mathcal{L}|,|\mathcal{S}|)$ | Mildew (31,4,0) | Mildew (31,0,4) | Mildew (29,3,3) | Water (29,3,0) | Water (29,0,3) | Water (26,3,3) |
|---|---|---|---|---|---|---|---|
| **L-MARVEL** | #CI tests | **359** | **194** | **426** | 2365 | **1130** | 1368 |
| | Runtime | **0.06** | **0.04** | **0.08** | **0.32** | 0.21 | 0.25 |
| | F1-score | **0.90** | **0.92** | **0.89** | **0.82** | **0.87** | **0.73** |
| | Precision | 0.95 | 0.99 | 0.96 | 0.97 | 0.98 | 0.95 |
| | Recall | **0.85** | **0.87** | **0.83** | **0.72** | **0.79** | **0.60** |
| **RFCI** | #CI tests | 896 | 1085 | 937 | 1472 | 1398 | 1173 |
| | Runtime | 0.20 | 0.23 | 0.19 | 0.21 | **0.29** | **0.22** |
| | F1-score | 0.77 | 0.84 | 0.79 | 0.67 | 0.69 | 0.60 |
| | Precision | **0.98** | 1.00 | 0.99 | 0.97 | 0.98 | 0.97 |
| | Recall | 0.64 | 0.73 | 0.66 | 0.51 | 0.53 | 0.44 |
| **FCI** | #CI tests | 1751 | 7251 | 10999 | 149674 | 12912 | 78903 |
| | Runtime | 0.33 | 1.57 | 2.26 | 29.99 | 2.99 | 19.00 |
| | F1-score | 0.72 | 0.81 | 0.74 | 0.57 | 0.61 | 0.50 |
| | Precision | **0.98** | **1.00** | **1.00** | 0.98 | 0.98 | 0.98 |
| | Recall | 0.57 | 0.69 | 0.59 | 0.41 | 0.45 | 0.34 |
| **MBCS*** | #CI tests | 1076 | 336 | 816 | 8300 | 3927 | 3946 |
| | Runtime | 0.28 | 0.12 | 0.25 | 1.98 | 1.07 | 1.12 |
| | F1-score | 0.81 | 0.89 | 0.82 | 0.68 | 0.74 | 0.61 |
| | Precision | 0.97 | 0.99 | 0.98 | **1.00** | **0.99** | **0.99** |
| | Recall | 0.70 | 0.81 | 0.71 | 0.52 | 0.59 | 0.45 |
| **M3HC** | #CI tests | 708 | 747 | 808 | **1591** | 1501 | **1285** |
| | Runtime | 8.41 | 9.93 | 17.33 | 36.65 | 78.99 | 61.48 |
| | F1-score | 0.76 | 0.79 | 0.75 | 0.65 | 0.63 | 0.57 |
| | Precision | **0.98** | **1.00** | 0.99 | 0.97 | 0.98 | 0.97 |
| | Recall | 0.62 | 0.66 | 0.61 | 0.48 | 0.47 | 0.40 |

## D   Specific excluded structure

In this section, we discuss the specific structure that is excluded from the result of Theorem 2. Formally, this structure is a MAG $\mathcal{M}$ that contains a specific type of cycle, which we call non-chordal: A cycle $(V_0, V_1, ..., V_m, V_{m+1} = V_0)$ such that I) $V_i$ and $V_{i+1}$ are neighbors for every $0 \leq i \leq m$, and II) the inducing subgraph of $\mathcal{M}$ over the vertices $\{V_0, ..., V_m\}$ does not contain any other edges. We show that this certain structure of MAGs represents a very restrictive structure of the DAG $\mathcal{G}_{\mathcal{V}}$. Consider the DAG $\mathcal{G}_{\mathcal{V}}$ in Figure 5a, where $\mathcal{O} = \{O_1, O_2, O_3, O_4\}$ and $\mathcal{S} = \{S_{12}, S_{23}, S_{34}, S_{41}\}$. The corresponding MAG is shown in Figure 5b. As seen in Figure 5b, the non-chordal cycle $(O_1, O_2, O_3, O_4, O_1)$ appears in the MAG structure. We claim such a cycle can only happen if all of the following conditions are satisfied:

- Each pair $(O_i, O_{i+1})$ have a specific selection variable $S_{i(i+1)}$ such that $O_i, O_{i+1} \in Anc(S_{i(i+1)})$, and none of the other observed variables of the cycle are ancestors of $S_{i(i+1)}$. Note that if for instance $O_1 \in Anc(S_{23})$ in the example above, then $O_1$ would be adjacent to $O_3$ in $\mathcal{G}_{\mathcal{V}}[\mathcal{O}|\mathcal{S}]$, since $(O_1, S_{23}, O_3)$ is an inducing path. So for the resulting MAG to have a non-chordal cycle, each pair of the observed variables must have their own specific selection variable.

- None of the pairs of variables $(O_i, O_j)$ must be adjacent if $j \neq (i-1), (i+1)$. That is, the induced subgraph of the DAG $\mathcal{G}_{\mathcal{V}}$ over $O_i$s must not contain any edges other than the edges of the cycle. Otherwise, the cycle in MAG $\mathcal{G}_{\mathcal{V}}[\mathcal{O}|\mathcal{S}]$ would contain a chord.

- None of the pairs of variables $(O_i, O_j)$ must have common latent confounders if $j \neq (i-1), (i+1)$. Otherwise, as in the case above, this would form a chord in the cycle.

Not allowing the aforementioned specific structure, the result of Theorem 2 is guaranteed. Note that it is mandatory to exclude this structure, as such structures have induced sub-graphs with no removable variables.

**Lemma 8.** *Suppose $\mathcal{G}$ is a MAG with non-chordal cycle $c = (O_0, ..., O_m)$. None of the vertices $\{O_0, ..., O_m\}$ are removable in any induced sub-graph of $\mathcal{G}$ that contains the cycle c.*

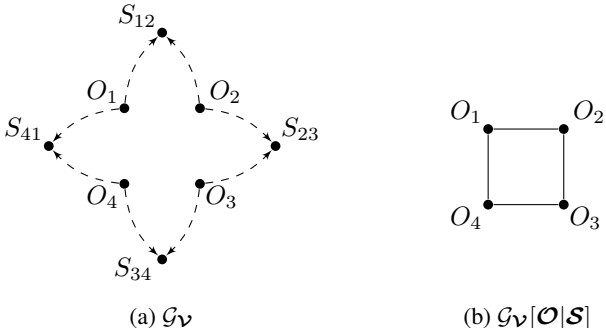

(a) $\mathcal{G_V}$  (b) $\mathcal{G_V}[\mathcal{O}|\mathcal{S}]$

Figure 5: A structure where every pair of observed vertices have its own specific selection variable, shared among only the variables of this pair. This results in a non-chordal MAG over the observe variables, if none of the pairs $(O_1, O_3)$ and $(O_2, O_4)$ have neither an edge in the DAG $\mathcal{G_V}$, nor a latent common confounder.

*Proof.* Suppose $\mathcal{H}$ is an induced sub-graph of $\mathcal{G}$ that contains the cycle $c$. Take an arbitrary vertex $O_i$. $O_{i-1}, O_{i+1} \in N(O_i)$, and $O_{i-1} \notin Adj(O_{i+1})$ since $c$ is non-chordal. From Theorem 1, $O_i$ is not removable, which completes the proof. $\qquad\square$

The following result indicates that given the aforementioned assumption, that is, if no non-chordal cycle exists in $\mathcal{M} = \mathcal{G_V}[\mathcal{O}|\mathcal{S}]$, then a removable variable always exists in any subgraph of $\mathcal{M}$, which completes our discussion.

**Lemma 9.** *Suppose the edge-induced subgraph of $\mathcal{M}$ over the undirected edges is chordal. Let $\mathcal{G} = \mathcal{G_V}[\mathbf{V}|\mathcal{S}]$ for some $\mathbf{V} \subseteq \mathcal{O}$. Then there exists $X \in \mathbf{V}$ such that $X$ is removable in $\mathcal{G}$.*

*Proof.* We consider the following two cases and introduce a removable variable at each case:

1. $\mathcal{G}$ has at least one directed or bidirected edge: Take $X$ as a vertex that has at least one arrowhead incident to it (i.e., it has at least a parent or a spouse), and satisfies the following property:
$$De_{\mathcal{G}}(X) \cap \mathbf{V} \setminus \{X\} = \varnothing.$$
We first show that such a vertex exists. Suppose not. Start from a vertex with an arrowhead incident to it and move to one of its children. Since the vertex we are in now has other descendants, again move to one of its children. Continuing in the same manner, we traverse over a directed cycle, which is in contradiction with the definition of MAGs.

Now we show that this variable $X$ is removable. Since $X$ has no other descendants, $Ch(X) = \varnothing$. Furthermore, $N(X) = \varnothing$ by definition of MAG. Now Theorem 1 implies that $X$ is removable.

2. $\mathcal{G}$ is an undirected graph: Since $\mathcal{M}$ is chordal over its undirected edges, $\mathcal{G}$ is chordal too. Chordal graphs have a perfect elimination ordering [7, 1]. Let $X$ be the first vertex in this ordering. By definition of perfect elimination ordering, all of the neighbors of $X$ are adjacent. From Theorem 1, $X$ is removable

$\square$

Lemmas 8 and 9 indicate that the assumption that the induced subgraph of $\mathcal{M}$ on the undirected edges is chordal is the necessary and sufficient condition so that there exists a removable variable at every subgraph of $\mathcal{M}$.