# OpenReview forum: "Recursive Causal Structure Learning in the Presence of Latent Variables and Selection Bias"
_NeurIPS.cc/2021/Conference — NeurIPS 2021 Poster_

### Official Review · Reviewer_pWDU · 2021-07-01

**Rating:** 7
**Confidence:** 3

**Summary:**

The paper presents a method to learn a Maximal Ancestral Graph (MAG), a mixed graph, an extension of DAGs, capable of representing a system with selection bias and hidden variables. The method is constraint-based (based on conditional independence tests) and recursive. The main innovation is the introduction of the concept of a “removable” variable, a variable that can be removed from the current graph such that the subgraph corresponding to the marginal distribution of the remaining variables can be learned correctly. The problem of learning the whole graph is then decomposed by always identifying the next removable variable and then recursively learning the graph for the remaining variables.

**Limitations And Societal Impact:**

Sufficiently yes, although the limitations are a bit narrowly discussed (just one sentence in Conclusion).

**Main Review:**

PROS:
- Very clear presentation. Also the math seems impeccable, although I only glanced through the lengthy proofs in the Appendix.
- Good motivation and intuition, and in general the problem is relevant to the community.
- Novel idea, which improves computational efficiency (because the removable variable has small Markov boundary resulting in a small number of CI tests in each iteration) and accuracy (because the order of the graph decreases over iterations and hence the size of the conditioning set in the CI tests is reduced).
- Theoretical analysis of the number of CI tests needed by the method and also a novel proof for the lower bound of the number of tests needed by any algorithm.
- Experiments with several synthetic and real graphs that thoroughly demonstrate that the method outperforms alternatives in terms of accuracy and/or runtime.

CONS:
- Actually, I did not identify any major limitations in this article (not with the first reading at least).

QUESTIONS:
- Could you clarify if there always is a removable variable when the size of V is greater than 1?
- Very minor: the output and input of the “isRemovable” function on line 8 in Algo1 are not the same or in the same order as in Algo2.



**Time Spent Reviewing:**

6

---

> ### Author Response · Authors · 2021-08-10
> **Response to Reviewer pWDU**
>
> We appreciate the comments and the points raised by the reviewer.
> ***
> Regarding the existence of a removable node in each iteration, as mentioned in Appendix D, there is only a very specific structure related to the edge induced subgraph of the MAG G over its undirected edges (i.e., those due to selection variables) which can result in G not having a removable variable: The requirement for having at least one removable variable is that the edge induced subgraph of G over its undirected edges is chordal. (We mentioned an excessive requirement in the appendix, which we later realised is not necessary.) Note that if this subgraph is chordal for the initial MAG, this condition holds for all the subsequent iterations. The violation of this requirement is very rare in the sense that it only happens if there is a cycle of length at least 4 consisting of only undirected edges, where each pair of adjacent variables on this cycle have a common selection variable which is not a descendant of any of the other variables on the cycle (Figure 5.a of the Appendix.) A proof of this result is as follows.
>
> - **Case1:** If there exists at least one directed or bidirected edge in the graph, then consider any node that has an arrowhead towards it on all of its incident edges (i.e, it has no children or neighbors. Such a node exists due to acyclicity of the MAG.) This node is removable due to Theorem 4 in the Appendix.
> - **Case 2:** If the graph consists only of undirected edges: By the chordal assumption, we know the graph has a perfect elimination ordering [A,B]. Now consider the first node in this ordering. The set of the neighbors of this node form a clique, and therefore it is removable (following from Theorem 4).
>
> [A] Fulkerson, Delbert, and Oliver Gross. "Incidence matrices and interval graphs." Pacific journal of mathematics 15.3 (1965): 835-855.
>
> [B] Blair, J. R., and Peyton, B. 1993. An introduction to chordal graphs and clique trees. In Graph theory and sparse matrix computation. Springer. 1–29.
> ***
> >  the output and input of the “isRemovable” function on line 8 in Algo1 are not the same or in the same order as in Algo2.
>
> We thank the reviewer for their preciseness. We will edit this in the final version.

---

> > ### Comment · Reviewer_pWDU · 2021-08-19
> > **Feedback after author response**
> >
> > Thanks for the clarifications. I will keep my original score.

---

### Official Review · Reviewer_QNSv · 2021-07-15

**Rating:** 7
**Confidence:** 2

**Summary:**

This paper focuses on causal discovery especially focusing on the casual structure learning in the presence of latent variables and selection bias. \
A recursive constraint-based method is proposed for learning causal structure. The Markov boundary information and 'removable' variable are introduced to reduce the CI tests.

**Limitations And Societal Impact:**

Yes

**Main Review:**

The paper considers an important problem of learning causal graphs in the presence of latent variables and selection bias.  \
The paper builds on constraint-based methods, such as FCI and RFCI.\
The paper provides a theoretical study of the connection between Markov boundary and separating set, which is closely related to the proposed algorithm. \
The analysis and the algorithm are sound and presented in a logical way.


However, there are some concerns or suggestions:
1. What's the reason for restricting the variables follow the structural equations model (Line 146)? It is not clear to me whether your method needs this assumption.  Please clarify it.

2. It's not clear to me whether the removable variables always exist in the subgraph?  It would be good to add some examples to explain the removable variable after definition 4 or Theorem 1.
3. In the experiments (Benchmark Structures Part), could you explain why the Precision of the RFCI is almost one while RLCD is not?

The algorithm's performance is a cause for concern, especially when the Markov boundary information can not discover correctly.
This is because the algorithm highly relies on the Markov boundary.

**Time Spent Reviewing:**

5 hours

---

> ### Author Response · Authors · 2021-08-10
> **Response to Reviewer QNSv**
>
> We thank the reviewer for their helpful comments and questions they raised.
>
> ***
> Regarding the SEM assumption in the paper, we need the assumption that the conditional independencies are equivalent to d-separations in the graph. To the best of our knowledge, the only two commonly used models in the literature that satisfy this property are the FFRCISTG model and the SEM model that we presented in the paper. Despite the fact that FFRCISTG is more general, since we are only working with observational data, these two models are equivalent for our application. Therefore, due to the fact that the used SEM model is easier to present and describe we used this model.
> ***
> > It would be good to add some examples to explain the removable variable after definition 4 or Theorem 1.
>
> We provided a graphical representation of a removable variable in appendix A, which we will discuss in the main text along with examples to give a clearer idea of the removable variables to the readers.
> ***
> Regarding the existence of a removable node in each iteration, as mentioned in Appendix D, there is only a very specific structure related to the edge induced subgraph of the MAG G over its undirected edges (i.e., those due to selection variables) which can result in G not having a removable variable: The requirement for having at least one removable variable is that the edge induced subgraph of G over its undirected edges is chordal. (We mentioned an excessive requirement in the appendix, which we later realised is not necessary.) Note that if this subgraph is chordal for the initial MAG, this condition holds for all the subsequent iterations. The violation of this requirement is very rare in the sense that it only happens if there is a cycle of length at least 4 consisting of only undirected edges, where each pair of adjacent variables on this cycle have a common selection variable which is not a descendant of any of the other variables on the cycle (Figure 5.a of the Appendix.) A proof of this result is as follows.
>
> - **Case1:** If there exists at least one directed or bidirected edge in the graph, then consider any node that has an arrowhead towards it on all of its incident edges (i.e, it has no children or neighbors. Such a node exists due to acyclicity of the MAG.) This node is removable due to Theorem 4 in the Appendix.
>
> - **Case 2:** If the graph consists only of undirected edges: By the chordal assumption, we know the graph has a perfect elimination ordering [A,B]. Now consider the first node in this ordering. The set of the neighbors of this node form a clique, and therefore it is removable (following from Theorem 4).
>
> [A] Fulkerson, Delbert, and Oliver Gross. "Incidence matrices and interval graphs." Pacific journal of mathematics 15.3 (1965): 835-855.
>
> [B] Blair, J. R., and Peyton, B. 1993. An introduction to chordal graphs and clique trees. In Graph theory and sparse matrix computation. Springer. 1–29.
> ***
> > In the experiments (Benchmark Structures Part), could you explain why the Precision of the RFCI is almost one while RLCD is not?
>
> One of the caveats of the constraint-based algorithms is that one single conditional independence is enough to remove an edge, i.e., they usually remove too many edges. As a result, constraint-based algorithms despite having high precision suffer from low recall in general, since they tend to remove edges too easily. RLCD avoids numerous CI tests by the virtue of its recursive nature, which means the chances of finding a CI relation among variables are reduced. This helps RLCD maintain a much higher recall at the expense of a slight loss in precision in practice. This is backed by our experiments where RLCD shows higher recall in almost every case, whereas it has slightly lower precision.

---

> > ### Comment · Reviewer_QNSv · 2021-08-18
> > **Feedback after author response**
> >
> > Thanks for the clarifications and for addressing my concerns. Based on the author's response and the authors addressing the concerns noted by all other reviewers, I have increased my score to a 7.

---

### Official Review · Reviewer_Wz2i · 2021-07-16

**Rating:** 7
**Confidence:** 4

**Summary:**

In this paper, the authors study the problem of learning a causal maximal ancestral graph (MAG) from observational data in the presence of latent variables and selection variables. They introduce a sound and complete recursive constraint-based algorithm (RLCD) with which they learn the causal structure efficiently. The algorithm uses Markov boundary information in order to reduce the number of required conditional independence (CI) tests. They introduce the notion of *removable* variables, which are variables that can be removed without changing the CI statements in the induced subgraph, and prove a graphical criterion for finding these variables from CI information. At each iteration of the algorithm, the *removable* variable with the smallest Markov boundary is removed, while the Markov boundaries and list of adjacencies are updated. The authors prove that this approach exhibits an upper bound on the number of CI tests that differs from a provable general lower bound only by (at most) a factor equal to the number of variables. The authors showcase the performance of their algorithm in a series of experiments on simulated data and benchmark Bayesian network structures.

**Limitations And Societal Impact:**

The authors have not addressed a technical limitation of their algorithm in the main paper, which involves excluding a specific restricted set of MAGs from the proof of Theorem 1, namely MAGs that contain non-chordal undirected cycles of length at least four. The authors have discussed this limitation adequately, however, in the supplement to the paper (see Remark 1 and Appendix D). Nevertheless, in the spirit of transparency, I think that this limitation should also be clearly mentioned in the main paper, even if just in footnote form.

**Main Review:**

*Originality*: The authors have come up with a very promising approach for learning the causal MAG from observational data by means of a sound and complete recursive algorithm through which the number of CI tests that have to performed is greatly reduced. While the idea of using *removable* variables has been previously employed for DAGs in reference [9] from the main paper, the extensions and proofs that the authors have introduced with this paper are, to the best of my knowledge, novel and greatly expand the applicability of the method by allowing for the presence of latent variables and selection bias.

*Quality*: The submission appears to be technically sound. I have partially checked the proofs in the supplement and the logic seems to hold water. The claims are well supported and the proofs are in general detailed enough.

A significant issue I have with the paper is that the proof of Theorem 1 excludes a very restricted set of MAGs, namely MAGs that contain non-chordal undirected cycles of length at least four. While I agree with the authors that this is a very restrictive structure and will not significantly influence the applicability of the algorithm, I do not find it completely above board that this caveat is only mentioned in the supplement. I think that the final version of the main paper should necessarily include this disclaimer, even if just as a footnote.

Another gripe I have is the definition for an $m$-connecting path, which I do not think is entirely accurate. The $m$-connecting part is fine, as under the conditions specified, $X$ and $Y$ will be $m$-connected. However, this connection may be due to a different path when the colliders are ancestors of the endpoints, namely a path that goes through the directed path from endpoint to collider.

*Clarity*: The paper is well structured and very well written, virtually error-free.

*Significance*: The results obtained are impressive and carry great significance. While constraint-best methods include some of the most tractable algorithms for causal discovery, they often run into trouble, particularly on dense graphs, when the number of conditional tests that have to be performed becomes exponential in the worst case, while the size of the conditioning tests performed becomes very large. This approach promises to tackle both of these issues to an important degree.

To further showcase the significance of the results, I would have also liked to see a comparison for sparse and dense graphs. Dense graphs are generally much more problematic for constraint-based causal inference than sparse graphs (see e.g., reference [2] from the paper), which is why I would have been really interested in looking at the benefit of the present work for the two cases separately. On that note, it would have been nice to include the FCI+ algorithm from the same reference [2] in the experiments, since it was specifically design to have a polynomial computational complexity on sparse graphs.

*Minor comments*:
-	Line 74: Formatting error – “Proposition 5” should be “Proposition 3”
-	Lines 201 and 206: I think $X$ should be $X_1$ everywhere.
-	Figure 2, rightmost column: y-axis label should be “Avg. Cond. Set Size”



**Time Spent Reviewing:**

six

---

> ### Author Response · Authors · 2021-08-10
> **Response to Reviewer Wz2i**
>
>  We appreciate the insightful comments of the reviewer. We are delighted that the reviewer finds our approach promising.
> ***
> Regarding Remark 1 in the appendix:
>
> - We will move the discussion of Remark 1 to the main text. Please note that we have mentioned an excessive assumption in the appendix, which we later realized is unnecessary. That is, given that the edge-induced subgraph over the undirected edges of the true MAG (those due to selection variables) is chordal, a removable variable always exists. Also, this assumption needs to hold only for the true MAG, as if a graph is chordal, any subgraph of it is chordal too, i.e., this assumption is satisfied throughout the algorithm if it only holds in the first iteration.
> ***
> Regarding the definition of m-separation:
>
> - Although it changes which path we consider as m-connecting, both of these definitions lead to the same result for the m-connection or m-separation of X and Y given a set Z. Since we only deal with m-separations and the exact path that m-connects two vertices are not of consequence in our work, we have opted to use this definition for ease of presentation.
> ***
> Regarding the comparison with FCI+:
>
> -  FCI+ requires the maximum degree ($\Delta$) as an input that cannot be estimated from data, unlike Markov boundaries required by our approach, which can be learned from data. Therefore, we have to resort to using an upper bound for $\Delta$ to be able to compare FCI+ with our approach. Note that the bound on the number of conditional independence tests proposed in that work is polynomial in $|V|$ only if the upper bound is a constant in the number of variables. Therefore, if we were to use the maximum size of the Markov boundary as a bound, as the reviewer suggests since this value is usually not constant in the number of variables, the resulting bound would not even be polynomial in the number of variables. Also, we note that if we use the maximum size of the Markov boundary as a bound, since Pa+ is always a subset of the Markov boundary, RLCD will achieve a much better bound than FCI+.

---

> > ### Comment · Reviewer_Wz2i · 2021-08-17
> > **Acknowledgment of response**
> >
> > Thank you for addressing my concerns with this response. I think this is a good paper, and my score remains unchanged.

---

### Official Review · Reviewer_SAUv · 2021-07-16

**Rating:** 7
**Confidence:** 4

**Summary:**

The authors propose a recursive structure learning algorithm when there are latent confounders and selection bias. They focus in on a subclass of graphs with a key property that there always is a vertex such that after its removal, the dependency structure of the remaining graph does not change.

The second key fact they utilize is that understanding if a node can be removed without changing the m-separation relations in the rest of the graph only requires the neighborhood information. This can be (somewhat efficiently) learned by first finding the Markov boundary of the node and then finding the separable nodes within the Markov boundary, which only requires testing subsets of the Markov boundary.

I did enjoy reading the paper overall and I believe it provides a good contribution.

**Limitations And Societal Impact:**

No. Limitations should be discussed. Only MB accuracy is mentioned as future work.

**Main Review:**

One issue that needs to be addressed is that the authors make an assumption about the graph structure to ensure this property is satisfied throughout. However, this assumption becomes apparent only in the appendix (proof of Thm. 2 and Lemma 10). I believe this assumption and node-removability from the MAG has to be the centerpiece of the discussion. I don't believe this is a very restrictive assumption, especially if there are no selection nodes, it should always hold (there is no partially directed cycle in a MAG by construction), however, the discussion should be there in the main paper.

Another issue is that Proposition 3 or (6), which gives the number of CI tests RCD uses excludes the initial phase of Markov boundary discovery. Please add this quadratic term to be precise. The discussion after Proposition 3 is useful.

The lower bound is useful but a discussion on this bound being over all graphs would be useful. An instance-wise bound will look very different from this.

Some comments on why p(V\{x}) is used instead of p(V\{x}|x) would be nice to have.

line 129: authors do not make the distinction between Ancestors and Anteriors of a node. This is because they only operate on a DAG rather than another ancestral graph. It might be worth making a footnote about this.

line 133: seems like the wrong reference as [26] doesn't include selection variables. Authors should refer to Richardson & Spirtes 2002 pg. 20.

I was initially a bit concerned that there is no explicit mention of the selection variables S in the proofs in Appendix A. I believe it's not needed since authors operate on a MAG level.

Lemma 6: Is G assumed to be a MAG?

Remark 1 of appendix, which is an assumption about the underlying graph, should be in the main text.

On Theorem 1: Suppose Z<->X-Y. Then Mb(X)={Y,Z}. Y is not ind of Z hence C1 fails. Similarly, Y is indep of Z given X then C2 fails. I believe this cannot be a MAG or a part of a MAG but examples like these would be very helpful to convey that there will always be a removable node, which is the crux of the approach.

theses variables-> these variables

line 760: FindAdjcent->FindAdjacent

Post-rebuttal: I would like to thank the authors for engaging in the discussion. The paper demonstrates a surprising fact about learning causal graphs - that a recursive approach taking out one node at a time is possible. I will keep my original score of "accept".

**Time Spent Reviewing:**

4

---

> ### Author Response · Authors · 2021-08-10
> **Response to Reviewer SAUv**
>
> We thank the reviewer for their helpful comments. Below we discuss the main points:
> ***
> Regarding the proof of Theorem 2 in the appendix:
>
> - We will move the discussion of Remark 1 to the main text. Please note that we have mentioned an excessive assumption in the appendix, which we later realized is unnecessary. That is, given that the edge-induced subgraph over the undirected edges of the true MAG (those due to selection variables) is chordal, a removable variable always exists. Also, this assumption needs to hold only for the true MAG, as if a graph is chordal, any subgraph of it is chordal too, i.e., this assumption is satisfied throughout the algorithm if it only holds in the first iteration.
>
> ***
> Regarding example for Theorem 1:
>
> - We will provide examples in the final version of our paper for the sake of more clarity. We have also provided a graphical representation of a removable variable in appendix A, which we will discuss in the main text to give readers a clearer idea of the removable variables. (The example here cannot be a part of a MAG as declared by the reviewer.)
>
> ***
> Regarding the quadratic term in Proposition 3:
>
> - We will add the quadratic term to our upper bound to be more precise and provide more discussion on the lower bound. We also thank the reviewer for other minor comments.
> ***
> > Some comments on why p(V{x}) is used instead of p(V{x}|x) would be nice to have.
>
> - We were not sure if we understood the reviewer’s point here.
> ***
> > Lemma 6: Is G assumed to be a MAG?
>
> G is assumed to be an undirected chordal graph, which is a specific type of MAG. There is a typo in the lemma, where it should be "If X and Z are not adjacent, then...". On a separate note, we found a more straightforward proof which skips a few of these lemmas:
>
> - **Case1:** If there exists at least one directed or bidirected edge in the graph, then consider any node that has an arrowhead towards it on all of its incident edges (i.e, it has no children or neighbors. Such a node exists due to acyclicity of the MAG.) This node is removable due to Theorem 4 in the Appendix.
> - **Case 2:** If the graph consists only of undirected edges: By the chordal assumption, we know the graph has a perfect elimination ordering [A,B]. Now consider the first node in this ordering. The set of the neighbors of this node form a clique, and therefore it is removable (following from Theorem 4).
>
> [A] Fulkerson, Delbert, and Oliver Gross. "Incidence matrices and interval graphs." Pacific journal of mathematics 15.3 (1965): 835-855.
>
> [B] Blair, J. R., and Peyton, B. 1993. An introduction to chordal graphs and clique trees. In Graph theory and sparse matrix computation. Springer. 1–29.

---

> > ### Comment · Reviewer_SAUv · 2021-08-16
> > **response**
> >
> > That's great. Thank you for the constructive, detailed responses.
> >
> > "p(V \backslash {x})" is converted to "p(V{x})" by openreview, I believe. What I meant was can you provide a high-level discussion on why the marginal on the remaining variables is used since one might also consider using the prob. of the rest of the observed variables given the variable that is just processed by the algorithm, i.e., p(V - {x}|x). Would such a step be more informative or no?

---

> > > ### Author Response · Authors · 2021-08-17
> > > **response**
> > >
> > > We thank the reviewer for the clarification. There are two reasons that we work with the marginal rather than the conditional distribution.
> > >
> > > First, conditioning on the processed variable $X$ is equivalent to $X$ being a selection variable, which would change the structure of the causal MAG over the remaining variables. However, we are trying to keep the graph structure invariant throughout the algorithm, so that we can learn it recursively.
> > >
> > > Second, marginalisation is easy in practice: we could just drop the values of the processed variable and use the same samples we have access to. However, if we wanted to work with conditionals, we would have to estimate them. Specially considering the fact that towards the end of algorithm we would have to estimate densities which are conditioned on too many variables (the set of all removed variables), this approach would end up with an impractical sample complexity.

---

> > > > ### Comment · Reviewer_SAUv · 2021-09-01
> > > > **response**
> > > >
> > > > Yes, that is what I was thinking about as well. It would be good to include this discussion in the camera-ready. Thank you for your reply.

---

### Decision · Program_Chairs · 2021-09-28

**Decision:**

Accept (Poster)

**Comment:**

This paper proposes a recursive structure learning algorithm for Maximal Ancestral Graph (MG) when there are latent variables and selection bias. The proposed algorithm is built on a novel notion of ``removal variable'' which many reviewers find interesting and reasonable. The theoretical analysis of computational efficiency and effectiveness was properly evaluated through experiments. All reviewers found the paper useful for the ML community. The authors are encouraged to consider updating the paper based on the reviewers' comments and suggestions.

**Consistency Experiment:**

NeurIPS has a long history of experimentation. In 2014, NeurIPS ran an experiment in which 10% of submissions were reviewed by two independent committees to quantify the randomness in the review process. This year, we repeated a variant of this experiment to see how the quality of the review process has changed over time.  This paper was part of the experiment and was therefore assigned to two committees (consisting of reviewers, an Area Chair, and a Senior Area Chair) that reached independent decisions.  If both committees made the same recommendation, this recommendation was followed. If a single committee recommended acceptance, the paper was accepted (with the exception of a few cases in which the other committee identified what we considered a fatal flaw, e.g., an error in a key result).

Both committees reached the same decision: **Accept (Poster)**

The other committee assigned to the paper recommended **Accept (Poster)**.  You can find the other set of reviews, along with any follow up discussion with the authors here:
https://openreview.net/forum?id=eElERAwRbo